# SimEdge: A Scalable Transitivity-Aware Graph-Theoretic Similarity Model for Capturing Edge-to-Edge Relationships

## ABSTRACT

Measuring similarity based on network topology is a crucial task in the realm of web search. While many well-established similarity measures (*e.g.* SimRank) focus on assessing node-to-node similarity, capturing edge-to-edge relationships is equally important in many applications (*e.g.* link spam detection). However, existing node-to-node similarity measures from the SimRank family may violate the triangular inequality. When applied directly to assessing edge-to-edge similarity, such measures may fail to capture transitive relationships and misrepresent dissimilarity between nodes.

In this paper, we propose a novel similarity measure, SimEdge, which can capture transitive relationships for assessing edge-to-edge similarity. The intuition of SimEdge revolves around a mutual reinforcement co-recursion: "two edges are assessed as similar if they are linked to similar nodes, and two nodes are assessed as similar if they are linked to similar edges." We show that SimEdge guarantees the transitivity of similarity, and enhances the accuracy of the node-to-node SimRank similarity without misrepresenting dissimilarity between nodes. For large-scale graphs, we also propose efficient techniques to compute SimEdge similarities in linear memory with guaranteed accuracy. Our empirical evaluation on various datasets validates that SimEdge is highly effective in capturing transitive edge-to-edge relationships, while offering a more reliable assessment of node-to-node similarity. Moreover, SimEdge shows superior scalability in assessing edge-to-edge similarities on large-scale graphs with billions of edges.

## 1 INTRODUCTION

Quantifying the similarity of two objects based on link structures is a fundamental problem in web search. Over the last decade, various link-based similarity measures have emerged for assessing node-to-node similarity [4, 5, 13, 14, 18, 20, 22]. Among them, the similarity models from the SimRank family (*e.g.* SimRank [12], CoSimRank [19], SimRank* [20], P-Rank [22]) have garnered growing attention due to their wide range of real-world applications, including web search, co-citation analysis, collaborative filtering, and social networks. The popularity of these measures is largely credited to two prominent features: 1) The core ideas underpinning these models are simple and intuitive, rendering their mathematical representations relatively easy for practical implementation. 2) The similarity scores generated by these models solely hinge on the underlying graph structures and can be iteratively propagated. As a result, the multi-hop neighborhood information between two nodes can be captured in a recursive and global manner.

However, these well-established similarity measures from the SimRank family are only designed for node-to-node similarity assessment. In practice, capturing edge-to-edge relationships in a graph holds equal importance in numerous real applications, such as neural synapse classification, link spam detection, co-citation analysis, fraud ring detection, and road network analysis.

APPLICATION 1 (NEURAL SYNAPSE CLASSIFICATION). *In a neural network, nodes represent neurons, and edges stand for synapses.*

*Different synapse types (e.g. Send-poly (Sp), Receive-poly (Rp), Electric Junction (EJ), Neuromuscular Junction (NMJ)) exhibit distinct connectivity patterns. Abnormal synaptic connectivity patterns are often associated with neurological disorders. Through edge-to-edge similarity assessment, we can distinguish these synapse types and identify unusual synaptic patterns associated with neurological disorders, aiding in the early diagnosis of neurological conditions.* □

APPLICATION 2 (LINK SPAM DETECTION). *Link spam often involves inserting links to irrelevant or low-quality websites within the content for the purpose of manipulating search engine rankings. Edge-to-edge similarity assessment based on hyperlinks in a web graph can help identify patterns and anomalies in the link structure of websites (e.g. link farms) by revealing web links with highly similar linking behaviors that lead to unrelated or spammy sites.* □

Despite its importance in many applications, effectively capturing the relationships between edges poses considerable challenges:

**Prior Approach.** A straightforward method to assess the similarity between edges in a graph $G$ is to construct a bipartite graph $G_B$ with two disjoint sets of nodes, $X$ and $Y$, where $X$ corresponds to all vertices in $G$, and $Y$ represents all edges in $G$. An edge $(x, y) \in X \times Y$ (*resp.* $(y, x) \in Y \times X$) exists in $G_B$ if $x$ is the outgoing (*resp.* incoming) edge of node $y$ in $G$. Then, any existing node-to-node similarity measure (*e.g.* [4, 13, 20–22]) can be directly applied to evaluate similarity in $G_B$. The similarity of any two nodes within the set $Y$ of $G_B$ corresponds to the similarity of two edges in $G$.

**Limitations.** However, there are two limitations to this approach:

Firstly, this method would significantly increase computational time for edge-to-edge similarity assessment, which does not scale on large graphs. To be specific, a conventional SimRank algorithm [20] applied to a graph $G$ with $|V|$ nodes and $|E|$ edges requires $O(|V||E|)$ time for evaluating similarities for $|V|^2$ pairs of nodes. Now, when constructing the bipartite graph $G_B$ from the original graph $G(V, E)$, we end up with two disjoint sets of nodes with a combined size of $(|V| + |E|)$ and a total of $2|E|$ edges in $G_B$. Consequently, if the existing SimRank algorithm [20] is directly applied to $G_B$, the time required for assessing similarities for $(|V| + |E|)^2$ pairs of nodes in $G_B$ — in order to obtain similarities for $|E|^2$ pairs of edges in $G$ — would be $O((|V| + |E|)|E|)$, which is prohibitively expensive.

Secondly, existing node-to-node similarity measures [4, 13, 20–22] from the SimRank family exhibit a lack of transitivity. To clarify, for any three nodes $a, b, c$ in a graph, if $a$ is similar to $b$ and $b$ is similar to $c$, it does not necessarily imply that $a$ is similar to $c$. This phenomenon arises due to the fact that these similarity measures are pseudometrics that may violate the triangular inequality. Consequently, when applied directly to assessing edge-to-edge similarity, these measures may struggle to capture transitive relationships in web search and may even lead to misrepresentations of dissimilarity between nodes (or edges), as illustrated by Example 1.

EXAMPLE 1. *Consider the web graph in Figure 1, where each node is a web page, and each edge is a hyperlink. Using SimRank measure, we can verify that the similarities among the three nodes $a, b, c$ are not transitive. Precisely, nodes $a$ and $b$ are similar ($sim(a, b) > 0$) since*

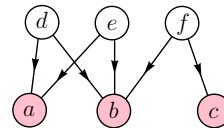

**Figure 1: SimRank and SimRank\* do not satisfy transitivity**

*they share two common in-neighbors $d$ and $e$. Similarly, nodes $b$ and $c$ are similar ($sim(b,c) > 0$) since they have a common in-neighbor $f$. However, nodes $a$ and $c$ are considered dissimilar ($sim(a,c) = 0$) by SimRank since there are no common 1-hop or multi-hop in-neighbors linking $a$ and $c$. This reveals that SimRank may not consistently exhibit transitivity since "$a \overset{sim}{\sim} b$ and $b \overset{sim}{\sim} c \nRightarrow a \overset{sim}{\sim} c$". When applied to edge-to-edge similarity analysis, this measure would fail to capture transitive relationships between edges as well. Worse, it misrepresents the dissimilarity between $a$ and $c$ ($sim(a,c) = 0$) despite the existence of several weakly connected paths (e.g. $a \leftarrow e \rightarrow b \leftarrow f \rightarrow c$).*    □

It is important to note that the transitivity problem of SimRank cannot be fully remedied by its variant models (*e.g.* SimRank\* [20], P-Rank [22], ASCOS++ [7]) or by simply adding self-loops to nodes. In Example 1, even if we replace the SimRank measure with SimRank\*, SimRank\* does not satisfy transitivity. This is because SimRank\* can only capture incoming paths with a single bifurcation node (*e.g.* $a \leftarrow \boxed{d} \rightarrow b$ with one bifurcation node $d$), and it still neglects all the weakly connected paths with multiple bifurcation nodes (*e.g.* $a \leftarrow \boxed{e} \rightarrow b \leftarrow \boxed{f} \rightarrow c$ with 2 bifurcation nodes $e$ and $f$). Consequently, using SimRank\* in Figure 1, we still have $sim(a,b) > 0$ and $sim(b,c) > 0$, but $sim(a,c) = 0$. This violates the transitivity.

The above example reveals that assessing edge-to-edge similarity in $G$ by simply treating all edges in $G$ as a separate set of nodes in its bipartite graph $G_B$ and then applying an existing node-to-node similarity model to $G_B$, is inadequate. Therefore, there is a pressing demand to devise an effective transitivity-aware model for edge-to-edge similarity assessment.

**Contributions.** The main contributions of this work are as follows:

- We propose a novel similarity model, SimEdge, based on a simple and intuitive mutual reinforcement co-recursion philosophy, which can effectively assess edge-to-edge similarity. (Section 3)
- We represent SimEdge in matrix forms and present a fixed-point iterative method to compute both edge-to-edge similarity and node-to-node similarity of SimEdge simultaneously. We also provide the error bounds for the SimEdge iterations. (Section 4)
- We theoretically justify that SimEdge can capture transitive relationships and fulfill triangular inequality. We also show how SimEdge avoids misrepresenting dissimilarity, and why it is more reliable than SimRank and SimRank\*. (Section 5)
- For large-scale graphs, we devise efficient techniques to substantially speed up the computation of SimEdge similarities within only linear memory, and propose a scalable algorithm for SimEdge search that is easy to parallelise over multi-core processors without any loss in accuracy. (Section 6)
- We conduct extensive experiments on various real-world datasets to demonstrate that (a) SimEdge achieves high accuracy (*e.g.* 0.97 MAP@100 and 0.95 NDCG@100 on DP) for quantifying edge-to-edge similarities. (b) In terms of accuracy for node-to-node similarities, SimEdge outperforms SimRank by a notable 19.3% and SimRank\* by 10.9%. (c) SimEdge shows superior scalability in assessing edge-to-edge similarities on large datasets with billions of edges. (Section 7)

## 2 RELATED WORK

Over the last decade, the majority of prior research has centered on assessing the similarity between nodes rather than edges in a graph. This inclination stems from the common belief among researchers that assessing edge-to-edge similarity in a graph can be tackled by simply applying the existing node-to-node similarity models (*e.g.* SimRank) to a bipartite graph with its two disjoint node sets corresponding to the vertex set and edge set of the original graph, respectively. However, this approach would introduce substantial computational overhead and may often result in low accuracy for edge-to-edge similarity, particularly when the underlying node-to-node similarity measure lacks transitivity, as shown in Section 1.

**SimRank.** SimRank, conceived by Jeh and Widom [12], is one of the most attractive graph-theoretic node-to-node similarity measures. However, SimRank exhibits some blemishes that may lead to undesirable similarities. These limitations include leakage of paths with odd lengths [7] and asymmetry [20], ignorance of out-links [22], over-relaxed normalisation factors [4, 8], dead loops in cycles [21], and indistinguishable self-similarity of all 1s [19].

To address these issues, some variants of SimRank have emerged, including SimRank\* [20], P-Rank [22], PSimRank [8], SimRank++ [4], CoSimRank [19], CoSimHeat [21], MatchSim [16], etc. However, one critical issue, plagued by SimRank but largely overlooked by these variant models, is the transitivity property.

**SimRank\* & P-Rank.** SimRank\* [20] provides a partial remedy by integrating asymmetric incoming paths between nodes that are ignored by SimRank. P-Rank [22] is another adaptation of SimRank that combines both in- and out-neighbors for assessing similarity, as opposed to SimRank which solely considers in-neighboring nodes. Nonetheless, neither SimRank\* nor P-Rank fulfil transitive property.

**PSimRank & SimRank++.** Fogaras *et al.* [8] and Antonellis *et al.* [4] observed the "connectivity trait" issue in SimRank, which leads to the counterintuitive decrease in similarity when there is an increase in the number of common in-neighbors between nodes. To resolve this issue, they introduced the PSimRank and SimRank++ models, respectively, by incorporating various weight factors into SimRank. However, these models do not ensure transitive property.

**CoSimRank & CoSimHeat.** Rothe and Schütze [19] proposed CoSimRank which excels in capturing all of the meeting time of two random surfers, unlike SimRank which only accounts for their first meeting time. However, there are "dead-loop" problems in cyclic graphs. When two random surfers enter a cycle, they endlessly chase each other and never meet again. As a consequence, all nodes on the cycle end up being evaluated as dissimilar. To circumvent this barrier, CoSimHeat [21] has been proposed recently. It leverages heat diffusion to mimic the activities of similarity propagation on the Web. Nonetheless, CoSimRank, like SimRank, lacks transitivity.

**RoleSim & MatchSim.** RoleSim [13] and MatchSim [16] incorporate "automorphism equivalence" into SimRank, based on the maximum weighted matching of the similarity between two in-neighbor sets. A slight disparity between RoleSim and MatchSim is the initialisation step. While MatchSim retains the initialisation of SimRank, whereas RoleSim initialises all-pairs scores with 1s. Both of these measures are based on role rather than distance.

**Edge Ranking & GNN.** There have also been studies on rating the importance of graph edges (*e.g.* [6, 10, 11]). Edge Betweenness Centrality (EBC) [6] is an appealing measure for ranking edges based on graph connectivity. EBC implies how the edges expedite

the flow of information in a graph. Node2Vec [10] is a graph embedding algorithm that converts a graph into a low-dimensional vector representation, which can also serve as an effective representation for edge features. Recently, Jana *et al.* [11] proposed a deep learning-based approach to estimate the EBC using a Graph Neural Network (GNN). Nevertheless, all of these studies concentrate on ranking edges rather than assessing the similarity between edges.

## 3 PROPOSED MODEL: SIMEDGE

**Notations.** Let $G = (V, E)$ be a digraph, where $V$ is the set of nodes, and $E$ is the set of edges. Nodes in $V$ are signified as $v_a, v_b, \cdots$, and edges in $E$ as $e_a, e_b, \cdots$. An edge originating from node $v_x$ and ending at $v_y$ can also be represented as an ordered pair of nodes $(v_x, v_y)$. We refer to $v_x$ as the *tail (node)* of the edge $e$, denoted as $T(e) = v_x$, and $v_y$ as the *head (node)* of $e$, denoted as $H(e) = v_y$.

The *in-link set* (*resp. out-link set*) of node $v$, denoted as $L^-(v)$ (*resp.* $L^+(v)$), is defined as the set of edges in the graph $G$ that have node $v$ as their head (*resp.* tail) node, that is,

$$L^-(v) = \{e \in E \mid H(e) = v\} \text{ and } L^+(v) = \{e \in E \mid T(e) = v\}$$

Throughout the paper, the following notations are adopted:
(a) Let $|X|$ be the cardinality of a set $X$. (b) We define

$$\underset{(x,y) \in X \times Y}{\text{average}} \{f(x,y)\} = \begin{cases} 0, & \text{if } X = \varnothing \text{ or } Y = \varnothing \\ \frac{1}{|X||Y|} \underset{(x,y) \in X \times Y}{\sum} f(x,y), & \text{otherwise.} \end{cases}$$

(c) $n = |V|$ (*resp.* $m = |E|$) is the number of nodes (*resp.* edges) in $G$.

**SimEdge Formulation.** The central theme behind the SimEdge measure is an intuitive mutual reinforcement co-recursion that

> "*two edges are similar if they are related to similar nodes;*
> *two nodes are similar if they are related to similar edges*".

The base case for this mutual co-recursion is to make each node and each edge most similar to itself.

Mathematically, the above ideas can be formulated as follows:

DEFINITION 1 (SIMEDGE). *Given digraph $G$ and damping factor $\gamma \in (0, 1)$, the SimEdge similarity between two edges $e_a$ and $e_b$ in $E$, denoted as $r(e_a, e_b)$, is defined by*

$$r(e_a, e_b) = \frac{\gamma}{4} \Big( s(H(e_a), H(e_b)) \tag{1a}$$
$$+ s(H(e_a), T(e_b)) \tag{1b}$$
$$+ s(T(e_a), H(e_b)) \tag{1c}$$
$$+ s(T(e_a), T(e_b)) \Big) \tag{1d}$$
$$+ (1 - \gamma) \cdot \begin{cases} 1, & \text{if } e_a = e_b; \\ 0, & \text{if } e_a \neq e_b. \end{cases} \tag{1e}$$

*where $s(*, *)$ is the SimEdge similarity score between nodes. For any two nodes $v_a$ and $v_b$ in $V$, $s(v_a, v_b)$ is defined by*

$$s(v_a, v_b) = \frac{\gamma}{4} \Big( \underset{(v_x, v_y) \in L^-(v_a) \times L^-(v_b)}{\text{average}} \{r(v_x, v_y)\} \tag{2a}$$
$$+ \underset{(v_x, v_y) \in L^-(v_a) \times L^+(v_b)}{\text{average}} \{r(v_x, v_y)\} \tag{2b}$$
$$+ \underset{(v_x, v_y) \in L^+(v_a) \times L^-(v_b)}{\text{average}} \{r(v_x, v_y)\} \tag{2c}$$
$$+ \underset{(v_x, v_y) \in L^+(v_a) \times L^+(v_b)}{\text{average}} \{r(v_x, v_y)\} \Big) \tag{2d}$$
$$+ (1 - \gamma) \cdot \begin{cases} 1, & \text{if } v_a = v_b; \\ 0, & \text{if } v_a \neq v_b. \end{cases} \tag{2e}$$

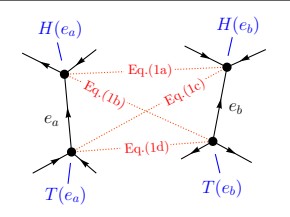
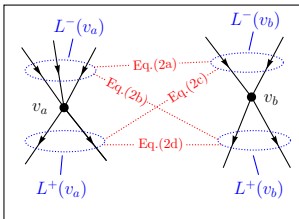

**Figure 2: Two edges $e_a$ and $e_b$ are similar if their related nodes are similar (dashed)**

**Figure 3: Two nodes $v_a$ and $v_b$ are similar if their related edges are similar (dashed)**

*Both edge-to-edge similarity $r(*, *)$ in Eq.(1) and node-to-node similarity $s(*, *)$ in Eq.(2) form a unified SimEdge measure.* □

Intuitively, edge-to-edge similarity $r(e_a, e_b)$ in Eq.(1) consists of five components. The first four components (1a)–(1d) are the average of its four related node-pair similarities between $e_a$'s head and $e_b$'s head (1a), $e_a$'s head and $e_b$'s tail (1b), $e_a$'s tail and $e_b$'s head (1c), $e_a$'s tail and $e_b$'s tail (1d). Figure 2 pictorially depicts the evaluation of these four node-pairs to quantify $r(e_a, e_b)$. The last component (1e) guarantees that every edge is most similar to itself. The choice of the factor $(1 - \gamma)$ in (1e) ensures all scores of $r(*, *)$ within $[0, 1]$. This is because, when $s(*, *) \in [0, 1]$, Eq.(1) implies

$$0 \leq r(e_a, e_b) \leq \frac{\gamma}{4}(1 + 1 + 1 + 1) + (1 - \gamma) \leq 1. \quad (\forall e_a, \forall e_b)$$

From $r(*, *) \in [0, 1]$, we can deduce via Eq.(2) that $s(*, *) \in [0, 1]$.

Analogously, node-to-node similarity $s(v_a, v_b)$ in Eq.(2) also consists of five components. The first four components (2a)–(2d) are four related averaged edge-pair similarities between $v_a$'s in- and $v_b$'s in-links (2a), $v_a$'s in- and $v_b$'s out-links (2b), $v_a$'s out- and $v_b$'s in-links (2c), $v_a$'s out- and $v_b$'s out-links (2d). Figure 3 picturizes the evaluation of these four sets of pairs of edges to measure $s(v_a, v_b)$. The last component (2e) guarantees that each node is most similar to itself, in which the factor $(1 - \gamma)$ ensures $s(*, *) \in [0, 1]$.

In our SimEdge model, edge-to-edge similarity $r(*, *)$ penetrates into its neighboring node-pairs, and recursively, node-to-node similarity $s(*, *)$ in turn penetrates into its neighboring edge-pairs. Thus, both edge- and node-pair similarities are mutually reinforced.

The damping factor $\gamma \in (0, 1)$ is a user-tuned parameter, which gives the rate of decay as similarity flows across edges and nodes. It is often set to 0.6–0.8, as previously used in SimRank [17].

## 4 MATRIX FORMULATION OF SIMEDGE

The SimEdge model in Definition 1 looks lengthy. In this section, we provide a concise matrix representation of SimEdge.

### 4.1 Matrix Form of SimEdge

**Forward & Backward Incidence Matrices.** To formulate SimEdge in matrix forms, we first introduce two incidence matrices.

DEFINITION 2. $\mathbf{A} \in \mathbb{R}^{n \times m}$ *is called the forward incidence matrix of a digraph $G = (V, E)$ whenever*

$$\mathbf{A}_{v_i, e_j} = \begin{cases} 1, & \text{if } v_i = T(e_j); \\ 0, & \text{otherwise.} \end{cases} \quad (\forall v_i \in V, \forall e_j \in E)$$

$\mathbf{B} \in \mathbb{R}^{n \times m}$ *is called the backward incidence matrix of $G$ whenever*

$$\mathbf{B}_{v_i, e_j} = \begin{cases} 1, & \text{if } v_i = H(e_j); \\ 0, & \text{otherwise.} \end{cases} \quad (\forall v_i \in V, \forall e_j \in E)$$

We also introduce matrices $\bar{\mathbf{A}}$ and $\bar{\mathbf{B}}$, which normalises all nonzero rows of $\mathbf{A}$ and $\mathbf{B}$, respectively, in the following manner:

$$\bar{\mathbf{A}}_{v_i,e_j} = \begin{cases} \frac{1}{|L^+(v_i)|}, & \text{if } v_i = T(e_j); \\ 0, & \text{otherwise.} \end{cases} \quad (\forall v_i \in V, \; \forall e_j \in E)$$

$$\bar{\mathbf{B}}_{v_i,e_j} = \begin{cases} \frac{1}{|L^-(v_i)|}, & \text{if } v_i = H(e_j); \\ 0, & \text{otherwise.} \end{cases} \quad (\forall v_i \in V, \; \forall e_j \in E)$$

**Co-resurive Form of SimEdge.** With forward and backward incidence matrices, we present the matrix representation of SimEdge.

THEOREM 1. *For digraph $G = (V, E)$, let $\mathbf{R} \in \mathbb{R}^{m \times m}$ and $\mathbf{S} \in \mathbb{R}^{n \times n}$ be its edge-to-edge and node-to-node similarity matrix, respectively, where $\mathbf{R}_{e_i,e_j} = r(e_i, e_j)$ and $\mathbf{S}_{v_i,v_j} = s(v_i, v_j)$. In matrix notations, the SimEdge model in Definition 1 can be expressed as*

$$\mathbf{R} = \gamma \cdot \mathbf{N}^T \mathbf{S} \mathbf{N} + (1 - \gamma) \cdot \mathbf{I}_m \tag{3a}$$

$$\mathbf{S} = \gamma \cdot \mathbf{M} \mathbf{R} \mathbf{M}^T + (1 - \gamma) \cdot \mathbf{I}_n \tag{3b}$$

*where $\mathbf{N}$ and $\mathbf{M}$ are two $n \times m$ matrices given by*

$$\mathbf{N} = \tfrac{1}{2}(\mathbf{A} + \mathbf{B}) \qquad \mathbf{M} = \tfrac{1}{2}(\bar{\mathbf{A}} + \bar{\mathbf{B}})$$

*and $\mathbf{I}_m \in \mathbb{R}^{m \times m}$ and $\mathbf{I}_n \in \mathbb{R}^{n \times n}$ are two identity matrices.* □

With this formulation, we will show the existence and uniqueness of SimEdge solution and derive the series form of SimEdge.

**Existence and Uniqueness.** Leveraging Theorem 1, we next prove the existence and uniqueness of SimEdge.

THEOREM 2. *There exists a unique solution for the edge-to-edge similarity $\mathbf{R}$ and node-to-node SimEdge similarity $\mathbf{S}$ in Eq.(3).* □

**Series Form of SimEdge.** Based on Theorems 1 and 2, we next get the power series form of the SimEdge solution $\mathbf{R}$ and $\mathbf{S}$ in Eq.(3).

THEOREM 3. *In the SimEdge model Eq.(3), the edge-to-edge similarity matrix $\mathbf{R}$ takes the following series form:*

$$\mathbf{R} = (1 - \gamma) \sum_{k=0}^{\infty} \gamma^{2k} \left(\mathbf{N}^T \mathbf{M}\right)^k \left(\gamma \mathbf{N}^T \mathbf{N} + \mathbf{I}_m\right) \left(\mathbf{M}^T \mathbf{N}\right)^k \tag{4}$$

*The node-to-node similarity matrix $\mathbf{S}$ can be expressed as*

$$\mathbf{S} = (1 - \gamma) \sum_{k=0}^{\infty} \gamma^{2k} \left(\mathbf{M} \mathbf{N}^T\right)^k \left(\gamma \mathbf{M} \mathbf{M}^T + \mathbf{I}_n\right) \left(\mathbf{N} \mathbf{M}^T\right)^k \tag{5}$$

Theorem 3 converts the mutual co-recursive form of SimEdge into two (infinite) matrix series. The convergence of these series in Eqs.(4) and (5) is guaranteed by Theorem 2. Moreover, with Theorem 3, we notice that $\mathbf{R} = \mathbf{R}^T$ and $\mathbf{S} = \mathbf{S}^T$. This implies that both edge-to-edge and node-to-node SimEdge matrices are symmetric.

## 4.2 Iteratively Computing $\mathbf{R}$ and $\mathbf{S}$

**Iterative Model of SimEdge.** Now that we have proved the existence and uniqueness of SimEdge, we next devise an iterative model capable of simultaneously obtaining $\mathbf{R}$ and $\mathbf{S}$ in Eq.(3).

THEOREM 4. *Let $\mathbf{R}_k$ and $\mathbf{S}_k$ be the edge-to-edge and node-to-node similarity matrices at the $k$-th iteration, respectively, obtained by*

$$\mathbf{S}_0 = (1 - \gamma) \cdot \mathbf{I}_n \tag{6a}$$

$$\mathbf{R}_k = \gamma \cdot \mathbf{N}^T \mathbf{S}_k \mathbf{N} + (1 - \gamma) \cdot \mathbf{I}_m \qquad (k = 0, 1, \cdots) \tag{6b}$$

$$\mathbf{S}_{k+1} = \gamma \cdot \mathbf{M} \mathbf{R}_k \mathbf{M}^T + (1 - \gamma) \cdot \mathbf{I}_n \qquad (k = 0, 1, \cdots) \tag{6c}$$

*After $l$ iterations, the following results hold:*

(a) $\mathbf{R}_l = \{\text{the first } l\text{-th partial sums of } \mathbf{R} \text{ in Eq.(4)}\}$;

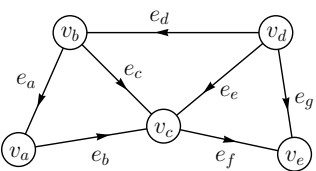

**Figure 4: Example of Computing SimEdge on Graph $G$**

(b) $\mathbf{S}_l = \{\text{the first } (l-1)\text{-th partial sums of } \mathbf{S} \text{ in Eq.(5)}\}$
$\quad + (1-\gamma)\gamma^{2l} \left(\mathbf{M} \mathbf{N}^T\right)^l \left(\mathbf{N} \mathbf{M}^T\right)^l$;

(c) $\mathbf{R}_l$ *(resp. $\mathbf{S}_l$) converges to $\mathbf{R}$ (resp. $\mathbf{S}$) as $l \to \infty$.*

Theorem 4 not only gives an iterative method to compute $\mathbf{R}$ and $\mathbf{S}$, but also discovers the relationship between the $l$-th iterative results in Eq.(6) and the first $l$-th partial sums of the series $\mathbf{R}$ and $\mathbf{S}$ in Eqs.(4) and (5). Unlike $\mathbf{R}_l$ that equals exactly the first $l$-th partial sums of Eq.(4), the value of $\mathbf{S}_l$ at iteration $l$ is between the first $(l-1)$-th and $l$-th partial sums of Eq.(5), but this does not affect the convergence of $\mathbf{S}_l$ to the same $\mathbf{S}$. This is because the gap $(1-\gamma)\gamma^{2l} \left(\mathbf{M} \mathbf{N}^T\right)^l \left(\mathbf{N} \mathbf{M}^T\right)^l$ between $\mathbf{S}_l$ and the first $(l-1)$-th partial sums of Eq.(5) converges to $\mathbf{0}$ as $l \to \infty$.

**Iterative Error Bound.** Capitalising on Theorem 4, we next analyze the accuracy of the iterative method in Eq.(6).

THEOREM 5. *Let $\mathbf{R}_l$ and $\mathbf{S}_l$ be the edge-to-edge and node-to-node similarity matrices at the $l$-th iteration of Eq.(6); $\mathbf{R}$ and $\mathbf{S}$ be the exact solutions. Then, for every iteration $l = 0, 1, \cdots$,*

$$\|\mathbf{R} - \mathbf{R}_l\|_{\max} \le \gamma^{2(l+1)} \quad and \quad \|\mathbf{S} - \mathbf{S}_l\|_{\max} \le \gamma^{2l}$$

*where $\|\mathbf{X}\|_{\max} = \max_{i,j} \{\mathbf{X}_{i,j}\}$ denotes the max norm of $\mathbf{X}$.* □

Theorem 5 gives a neat a-priori upper bound on the gap between the $l$-th iterative and exact similarities. It implies that, for attaining a desirable accuracy $\epsilon > 0$, the number of iterations required to compute $\mathbf{R}_l$ and $\mathbf{S}_l$ is $\lceil \frac{1}{2} \log_\gamma \epsilon \rceil$ and $\lceil \frac{1}{2} \log_\gamma \epsilon \rceil + 1$, respectively.

EXAMPLE 2. *Consider the network $G$ in Figure 4. Given damping factor $\gamma = 0.6$ and desirable accuracy $\epsilon = 0.001$, we compute edge-to-edge similarity matrix $\mathbf{R}$ and node-to-node similarity matrix $\mathbf{S}$.*

*First, based on the forward and backward incidence matrices $\mathbf{A}$ and $\mathbf{B}$ in Definition 2, $\mathbf{N}$ and $\mathbf{M}$ can be derived by Theorem 1 as follows:*

$$\mathbf{N} = \begin{array}{c|ccccccc} & e_a & e_b & e_c & e_d & e_e & e_f & e_g \\ \hline v_a & \frac{1}{2} & \frac{1}{2} & 0 & 0 & 0 & 0 & 0 \\ v_b & \frac{1}{2} & 0 & \frac{1}{2} & \frac{1}{2} & 0 & 0 & 0 \\ v_c & 0 & \frac{1}{2} & \frac{1}{2} & 0 & \frac{1}{2} & \frac{1}{2} & 0 \\ v_d & 0 & 0 & 0 & \frac{1}{2} & \frac{1}{2} & 0 & \frac{1}{2} \\ v_e & 0 & 0 & 0 & 0 & 0 & \frac{1}{2} & \frac{1}{2} \end{array}$$

$$\mathbf{M} = \begin{array}{c|ccccccc} & e_a & e_b & e_c & e_d & e_e & e_f & e_g \\ \hline v_a & \frac{1}{2} & \frac{1}{2} & 0 & 0 & 0 & 0 & 0 \\ v_b & \frac{1}{4} & 0 & \frac{1}{4} & \frac{1}{2} & 0 & 0 & 0 \\ v_c & 0 & \frac{1}{6} & \frac{1}{6} & 0 & \frac{1}{6} & \frac{1}{2} & 0 \\ v_d & 0 & 0 & 0 & \frac{1}{6} & \frac{1}{6} & 0 & \frac{1}{6} \\ v_e & 0 & 0 & 0 & 0 & 0 & \frac{1}{4} & \frac{1}{4} \end{array}$$

*Applying Theorem 5, we obtain the number of iterations required for computing $\mathbf{R}_l$ and $\mathbf{S}_l$ is, respectively,*

$$\lceil \tfrac{1}{2} \log_\gamma \epsilon \rceil = \lceil \tfrac{1}{2} \log_{0.6} 0.001 \rceil = 6 \; and \; \lceil \tfrac{1}{2} \log_\gamma \epsilon \rceil + 1 = 7.$$

*By Theorem 4, $\mathbf{R}_6$ and $\mathbf{S}_7$ can be iteratively computed as*

$$\mathbf{R}_6 = \begin{array}{c|ccccccc} & e_a & e_b & e_c & e_d & e_e & e_f & e_g \\ \hline e_a & .606 & .125 & .118 & .109 & .028 & .022 & .013 \\ e_b & .125 & .598 & .116 & .028 & .101 & .105 & .017 \\ e_c & .118 & .116 & .584 & .104 & .102 & .102 & .022 \\ e_d & .109 & .028 & .104 & .565 & .084 & .017 & .078 \\ e_e & .028 & .101 & .102 & .084 & .557 & .100 & .082 \\ e_f & .022 & .105 & .102 & .017 & .100 & .568 & .084 \\ e_g & .013 & .017 & .022 & .078 & .082 & .084 & .539 \end{array}$$

$$\mathbf{S}_7 = \begin{array}{c|ccccc} & v_a & v_b & v_c & v_d & v_e \\ \hline v_a & .618 & .093 & .073 & .015 & .012 \\ v_b & .093 & .570 & .049 & .046 & .013 \\ v_c & .073 & .049 & .556 & .028 & .060 \\ v_d & .015 & .046 & .028 & .436 & .023 \\ v_e & .012 & .013 & .060 & .023 & .448 \end{array} \quad \square$$

## 5 MEANINGFUL SEMANTICS OF SIMEDGE

In this section, we will theoretically substantiate two key points: 1) the effectiveness of SimEdge in capturing transitive relationships by maintaining the triangular inequality, and 2) how SimEdge avoids misrepresenting dissimilarity, making its node-to-node similarity $\mathbf{S}$ more meaningful than SimRank and SimRank*.

**Transitivity of SimEdge.** To show that SimEdge is capable of capturing transitive relationships, we will begin by presenting two lemmas. These lemmas will serve as the groundwork for the subsequent proof of SimEdge triangular inequality.

LEMMA 1. *For every iteration number $k = 0, 1, 2, \cdots$, if $\mathbf{R}_k$ in Eq.(6c) satisfies*

$$[\mathbf{R}_k]_{e_a,e_c} - [\mathbf{R}_k]_{e_a,e_b} \geq [\mathbf{R}_k]_{e_b,e_c} - 1, \quad (\forall e_a, e_b, e_c \in E)$$

*then, for $\mathbf{S}_{k+1}$ in Eq.(6c), the following inequality holds:*

$$[\mathbf{S}_{k+1}]_{v_a,v_c} - [\mathbf{S}_{k+1}]_{v_a,v_b} \geq [\mathbf{S}_{k+1}]_{v_b,v_c} - 1. \quad (\forall v_a, v_b, v_c \in V) \quad \square$$

Lemma 1 indicates that, for the SimEdge model, when the node-to-node similarity at the current iteration satisfies the triangular inequality, the edge-to-edge similarity in the next iteration also complies with the triangular inequality.

LEMMA 2. *For every iteration number $k = 0, 1, 2, \cdots$, if $\mathbf{S}_k$ in Eq.(6b) satisfies*

$$[\mathbf{S}_k]_{v_a,v_c} - [\mathbf{S}_k]_{v_a,v_b} \geq [\mathbf{S}_k]_{v_b,v_c} - 1. \quad (\forall v_a, v_b, v_c \in V)$$

*then, for $\mathbf{R}_k$ in Eq.(6b), the following inequality holds:*

$$[\mathbf{R}_k]_{e_a,e_c} - [\mathbf{R}_k]_{e_a,e_b} \geq [\mathbf{R}_k]_{e_b,e_c} - 1, \quad (\forall e_a, e_b, e_c \in E) \quad \square$$

Lemma 2 suggests that, at each current iteration, when the edge-to-edge similarity fulfills the triangular inequality, the node-to-node similarity also adheres to the triangular inequality.

Combining Lemmas 1 and 2, we observe that the adherence to the triangular inequality will persist in its transmission from the node layer to the edge layer in each iteration and then from the edge layer back to the node layer in the subsequent iteration, mutually reinforcing this property. Building upon these two lemmas, we next show that the SimEdge measure satisfies the triangular inequality. This result ensures the transitive property of its similarity values.

THEOREM 6. *For any two nodes $v_a$ and $v_b$, let $s(v_a, v_b)$ be their node-to-node SimEdge similarity. We define*

$$dist_s(v_a, v_b) \triangleq 1 - s(v_a, v_b)$$

*to be the SimEdge closeness between nodes $v_a$ and $v_b$. Then,*

(a) $dist_s(v_a, v_b) \geq 0$;
(b) $dist_s(v_a, v_b) = dist_s(v_b, v_a)$;
(c) $dist_s(v_a, v_c) \leq dist_s(v_a, v_b) + dist_s(v_b, v_c) \quad \forall v_a, v_b, v_c. \quad \square$

It is worth noting that the importance of the transitive property in SimEdge lies in its role in maintaining consistent similarity propagation within a network. Moreover, combined with Lemma 2, Theorem 6 implies that the transitivity for edge-to-edge similarity also holds. In comparison, other existing similarity measures from the SimRank family (*e.g.* SimRank*, P-Rank) lack transitivity, which can lead to counterintuitive results (see Example 1 in Section 1).

**Comparison with Other Similarity Measures.** In light of the transitive property of SimEdge, we next elucidate how SimEdge enhances the accuracy of SimRank and other variants (*e.g.* SimRank*).

Recently, some pioneering studies have observed the limitations of the SimRank measure – it may yield undesired similarities due to its failure to account for 1) paths with odd lengths [7], 2) asymmetric paths between nodes [20], and 3) cycles in graphs that lead to dead-loops [21]. To address these limitations, several promising variant models have been proposed [7, 20, 21]. Among them, SimRank* [20] stands out as a notable solution that allows SimRank to capture asymmetric paths without significantly increasing computational cost. These newly included paths by SimRank* encompass those in other models [7, 21] to enhance the SimRank accuracy. However, SimRank* still suffers from some limitations, as shown below:

THEOREM 7. *To quantify the node-to-node similarity $s(v_a, v_b)$, SimRank* [20] captures only the limited weakly connected paths of length $k$ ($\forall k = 1, 2, \cdots$) between nodes $v_a$ and $v_b$:*

$$(v_a =)v_0, e_0, v_1, e_1, v_2, \cdots, v_{k-1}, e_{k-1}, v_k(= v_b) \quad (7)$$

*under the constraints that, for any fixed $\alpha \in \{0, 1, \cdots, k-1\}$, its first $\alpha$ edges $\{e_0, e_1, \cdots, e_{\alpha-1}\}$ bear "$\leftarrow$" directions and last $(k - \alpha)$ edges $\{e_\alpha, e_{\alpha+1}, \cdots, e_{k-1}\}$ bear "$\rightarrow$" directions.* $\square$

Theorem 7 indicates that SimRank* can only partially resolve the "zero-similarity" problem in SimRank, as it still overlooks many weakly connected paths for similarity assessment. In contrast, these disregarded paths in SimRank* can be effectively accommodated by SimEdge due to its transitive capability to intertwine edge-to-edge relationships with its node-to-node similarity, as indicated below.

THEOREM 8. *SimEdge (Eq.(2)) can comprehensively quantify node-to-node similarity $s(v_a, v_b)$ by capturing all paths (including weakly connected paths) between nodes $v_a$ and $v_b$.* $\square$

EXAMPLE 3. *Let us consider the following 12 weakly connected paths of length 4 between two nodes $v_a$ and $v_b$.*

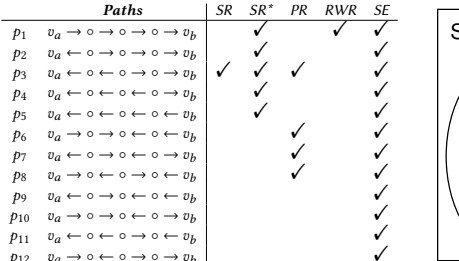
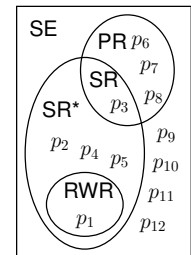

| Paths | | SR | SR* | PR | RWR | SE |
|---|---|---|---|---|---|---|
| $p_1$ | $v_a \rightarrow \circ \rightarrow \circ \rightarrow \circ \rightarrow v_b$ | | ✓ | | ✓ | ✓ |
| $p_2$ | $v_a \leftarrow \circ \rightarrow \circ \rightarrow \circ \rightarrow v_b$ | | ✓ | | | ✓ |
| $p_3$ | $v_a \leftarrow \circ \leftarrow \circ \rightarrow \circ \rightarrow v_b$ | ✓ | ✓ | ✓ | | ✓ |
| $p_4$ | $v_a \leftarrow \circ \leftarrow \circ \leftarrow \circ \rightarrow v_b$ | | ✓ | | | ✓ |
| $p_5$ | $v_a \leftarrow \circ \leftarrow \circ \leftarrow \circ \leftarrow v_b$ | | ✓ | | | ✓ |
| $p_6$ | $v_a \rightarrow \circ \rightarrow \circ \rightarrow \circ \leftarrow v_b$ | | | ✓ | | ✓ |
| $p_7$ | $v_a \leftarrow \circ \rightarrow \circ \rightarrow \circ \leftarrow v_b$ | | | ✓ | | ✓ |
| $p_8$ | $v_a \rightarrow \circ \leftarrow \circ \rightarrow \circ \leftarrow v_b$ | | | ✓ | | ✓ |
| $p_9$ | $v_a \leftarrow \circ \rightarrow \circ \leftarrow \circ \leftarrow v_b$ | | | | | ✓ |
| $p_{10}$ | $v_a \rightarrow \circ \rightarrow \circ \leftarrow \circ \rightarrow v_b$ | | | | | ✓ |
| $p_{11}$ | $v_a \leftarrow \circ \leftarrow \circ \rightarrow \circ \leftarrow v_b$ | | | | | ✓ |
| $p_{12}$ | $v_a \rightarrow \circ \leftarrow \circ \rightarrow \circ \rightarrow v_b$ | | | | | ✓ |

*To quantify $s(v_a, v_b)$, we have the following:*

(a) *SimRank (SR) captures only one path ($p_3$), whose single bifurcation node is in the center of the path.*
(b) *SimRank* (SR*) captures 5 paths ($p_1$) − ($p_5$), whose single bifurcation node can be at any position of the path, but with all edges on its left (resp. right) side bearing "$\leftarrow$" (resp. "$\rightarrow$") directions.*
(c) *P-Rank (PR) captures only 4 paths ($p_3$), ($p_6$), ($p_7$), ($p_8$), whose edge directions are symmetric w.r.t. the center node of the path.*
(d) *Random Walk with Restart (RWR) captures only one unidirectional path ($p_1$) from $v_a$ to $v_b$.*
(e) *SimEdge (SE) captures all these weakly connected paths, most of which are ignored by SimRank and SimRank*.* $\square$

## 6 FAST SIMEDGE COMPUTATION AT SCALE

In this section, we propose efficient scalable techniques to compute SimEdge similarities on large graphs using only linear memory.

Recall the iterative algorithm of SimEdge in Theorem 4 (Section 4.2). There are several limitations that hinder its scalability to large graphs: (a) The memory required for iteratively computing $\mathbf{R}_k$ and $\mathbf{S}_k$ by Eq.(6) is quadratic ($O(m^2)$). (b) All entries of $\mathbf{R}_k$ (resp. $\mathbf{S}_k$)

**Algorithm 1:** Evaluate Edge-To-Edge SimEdge Similarities

**Input** : digraph $G = (V, E)$, query edge $e_i \in E$,
damping factor $\gamma$, desired accuracy $\epsilon$.
**Output**: edge-to-edge SimEdge similarities $[\mathbf{R}]_{\star, e_i}$
between all edges in $G$ and query $e_i$.

1 build incidence matrices $\mathbf{A}, \mathbf{B}, \bar{\mathbf{A}}, \bar{\mathbf{B}}$ by Definition 2 ;
2 initialize $\mathbf{N} \leftarrow \frac{1}{2}(\mathbf{A} + \mathbf{B})$ and $\mathbf{M} \leftarrow \frac{1}{2}(\bar{\mathbf{A}} + \bar{\mathbf{B}})$ ;
3 determine the number of iterations $k \leftarrow \lceil \frac{1}{2} \log_\gamma \epsilon \rceil$ ;
4 initialize $\xi_0 \leftarrow [\mathbf{I}_m]_{\star, e_i}$ ;
5 **for** $l \leftarrow 1, 2, \cdots, k$ **do**
6      update $\xi_l \leftarrow \gamma \mathbf{M}^T (\mathbf{N}\xi_{l-1})$ ;
7 initialize $\eta_0 \leftarrow \xi_k + \gamma \mathbf{N}^T (\mathbf{N}\xi_k)$ ;
8 **for** $l \leftarrow 1, 2, \cdots, k$ **do**
9      update $\eta_l \leftarrow \xi_{k-l} + \gamma \mathbf{N}^T (\mathbf{N}\xi_{k-l} + \mathbf{M}\eta_{l-1})$ ;
10 **return** $[\mathbf{R}_k]_{\star, e_i} \leftarrow (1 - \gamma)\eta_k$ ;

in Eq.(6) have to be evaluated simultaneously even if some applications may necessitate only a subset of elements within $\mathbf{R}_k$ (resp. $\mathbf{S}_k$). (c) Eq.(6) disallows the independent evaluation of edge-to-edge and node-to-node similarities, even though in some cases, users may only require edge-to-edge similarities. To address these issues, we next propose an efficient method to compute $\mathbf{R}_k$.

THEOREM 9. *Given a digraph $G$, a query edge $e_i$, and the number of iterations $k$, the SimEdge similarities $[\mathbf{R}_k]_{\star, e_i}$ between every edge in $G$ and query $e_i$ at the $k$-th iteration of Eq.(6b) can be computed as*

$$[\mathbf{R}_k]_{\star, e_i} = (1 - \gamma)\eta_k$$

*where auxiliary vector $\eta_k$ can be iteratively reached as*

$$\begin{cases} \eta_0 = \xi_k + \gamma \mathbf{N}^T (\mathbf{N}\xi_k) \\ \eta_l = \xi_{k-l} + \gamma \mathbf{N}^T (\mathbf{N}\xi_{k-l} + \mathbf{M}\eta_{l-1}) \quad (l = 1, \cdots, k) \end{cases} \quad (8)$$

*with auxiliary vectors $\xi_0, \cdots, \xi_k$ iteratively obtained by*

$$\begin{cases} \xi_0 = [\mathbf{I}_m]_{\star, e_i} \\ \xi_l = \gamma \mathbf{M}^T (\mathbf{N}\xi_{l-1}) \quad (l = 1, \cdots, k) \end{cases} \quad (9)$$

**Algorithm.** Theorem 9 provides an efficient way to compute $\mathbf{R}_k$ column by column, independently, in linear memory, as shown in Algorithm 1. Initially, the algorithm first manages two sparse matrices $\mathbf{N}$ and $\mathbf{M}$. Next, it iteratively obtains $k$ auxiliary vectors $\xi_0, \cdots, \xi_k$, and then computes $\eta_k$, whose scaled result is $[\mathbf{R}]_{\star, e_i}$.

**Time & Space Complexity.** The computational time and space complexities of Algorithm 1 are analysed as follows.

THEOREM 10. *Given a query edge $e_i$ and the total number of iterations $k$, it requires $O(km)$ time and $O(km)$ memory to compute $[\mathbf{R}_k]_{\star, e_i} \in \mathbb{R}^{m \times 1}$ via Algorithm 1.* □

Compared with our previous model in Eq.(6), Algorithm 1 highlights the following advantages: (a) Each column of $\mathbf{R}_k$ can be computed independently and in parallel as needed, without relying on $\mathbf{S}_k$ or other columns of $\mathbf{R}_k$. (b) The memory required for computing all pairs of $\mathbf{R}_k$ is reduced from $O(m^2)$ of Eq.(6) to $O(km)$ $(k \ll m)$[1].
**Correctness.** The correctness of the edge-to-edge similarity results returned by Algorithm 1 is guaranteed by Theorem 9.

EXAMPLE 4. *Recall Example 2 and $G$ in Figure 4. Given $\gamma = 0.6$ and $k = 6$, we show how Theorem 9 computes only the SimEdge similarities between all edges in $G$ and query edge $e_5$.*

---

[1]For real large networks, $m$ is $10^5 \sim 10^7$, whereas $k \approx 20$ $(\ll m)$.

*First, we employ Eqs.(8)–(9) to iteratively get $\xi_k$ and $\eta_k$:*

| $k$ | $\xi_k$ | $\eta_k$ |
|---|---|---|
| 0 | $[0\ \ 0\ \ 0\ \ 1\ \ 0\ \ 0]^T$ | $[.003\ .003\ .003\ .003\ .004\ .002]^T$ |
| 1 | $[0\ \ .050\ .050\ .050\ .100\ .150\ .050]^T$ | $[.008\ .009\ .009\ .008\ .007\ .011\ .006]^T$ |
| 2 | $[.015\ .025\ .025\ .025\ .028\ .068\ .025]^T$ | $[.016\ .019\ .019\ .016\ .016\ .024\ .012]^T$ |
| 3 | $[.011\ .013\ .012\ .014\ .011\ .029\ .011]^T$ | $[.030\ .038\ .039\ .032\ .035\ .053\ .027]^T$ |
| 4 | $[.006\ .007\ .006\ .007\ .005\ .013\ .005]^T$ | $[.051\ .077\ .080\ .061\ .079\ .123\ .061]^T$ |
| 5 | $[.004\ .004\ .003\ .004\ .002\ .006\ .002]^T$ | $[.061\ .159\ .167\ .124\ .223\ .277\ .134]^T$ |
| 6 | $[.002\ .002\ .002\ .002\ .001\ .003\ .001]^T$ | $[.069\ .252\ .255\ .210\ .393\ .250\ .205]^T$ |

*Then, leveraging $\eta_6$ and $\gamma = 0.6$, we can obtain*

$$[\mathbf{R}_6]_{\star, e_5} = (1 - 0.6)\eta_6 = [.028\ .101\ .102\ .084\ .557\ .100\ .082]^T \quad \square$$

**Error Bound.** Regarding the accuracy of Algorithm 2, a method similar to Theorem 5 can be applied to show that, for $\forall e_i \in E$,

$$\|[\mathbf{R}]_{\star, e_i} - [\mathbf{R}_k]_{\star, e_i}\|_{\max} \leq \gamma^{2(k+1)} \quad (k = 1, 2, \cdots)$$

where $[\mathbf{R}]_{\star, e_i}$ is the column $e_i$ of the exact solution $\mathbf{R}$, and $[\mathbf{R}_k]_{\star, e_i}$ is the resulting vector at the $k$-th iteration returned by Algorithm 1.
**Extension to Node-To-Node Similarity Assessment.** Analogous to Theorems 9–10, a similar efficient algorithm can be devised to evaluate node-to-node SimEdge similarities $\mathbf{S}_k$ of Eq.(6) column by column independently and in parallel, with just $O(m + kn)$ linear memory for $k$ iterations (see Algorithm 2 in Appendix B).

## 7 EXPERIMENTAL STUDY

We will validate: (a) the high accuracy of SimEdge to quantify edge-to-edge relationships; (b) more meaningful node-to-node semantics of SimEdge; and (c) its fast speed and scalability on large graphs.

### 7.1 Experimental Settings

**(1) Real-World Datasets.** The datasets are described as follows:

| Data | $|E|$ | $|V|$ | $|E|/|V|$ | Scale | Description |
|---|---|---|---|---|---|
| NU | 3,677 | 182 | 20.2 | small | neuronal connectivity network |
| DP | 27,550 | 5,001 | 5.5 | small | co-authorship DBLP network |
| AL | 67,663 | 3,425 | 19.8 | small | snapshot of OpenFlights network |
| HP | 118,521 | 12,008 | 9.9 | medium | citation network from arXiv HEP |
| AM | 925,872 | 334,863 | 2.8 | medium | Amazon co-purchasing network |
| LJ | 34,681,189 | 3,997,962 | 8.7 | large | LiveJournal online social network |
| EU | 386,915,963 | 11,264,052 | 16.1 | large | EU web hosting infrastructure |
| TW | 1,468,365,182 | 41,652,230 | 35.3 | large | large-scale social network on Twitter |

The detailed description of each dataset is given in Appendix C.1.
**(2) Synthetic Datasets.** We use GTgraph [2], a synthetic graph generator that takes as input the number of nodes ($n$) and edges ($m$), and adds $m$ edges randomly each time.
**(3) Query Sampling.** To achieve statistical significance, when assessing edge-to-edge similarities $r(*, q)$ *w.r.t.* a given edge query $q$, we randomly choose 300 queries from $E$ as follows: We first calculate the PageRank values of every node in $V$, and then assign a weight $w(e) = \frac{1}{2}(\text{PR}(H(e)) + \text{PR}(T(e)))$ to each edge $e$ in $E$, where $\text{PR}(H(e))$ is the PageRank value of $e$'s head node, and $\text{PR}(T(e))$ is that of $e$'s tail node. To ensure a comprehensive coverage of potential queries, we rank all edges in $E$ based on their weight $w(e)$ and divide them into 5 buckets. The first bucket comprises edges with $w(e)$ in the range $[0.8, 1]$, followed by the second bucket with edges in the range $[0.6, 0.8)$, and so on. Subsequently, we randomly pick 60 edges from each bucket, guaranteeing that all 300 queries encompass both significant and less significant edges.

Similarly, to assess node-to-node similarity $s(*, v)$, we randomly choose 300 queries from $V$ based on the PageRank value $\text{PR}(v)$.
**(4) Evaluation Metrics.** Two metrics are used to measure accuracy: MAP (Mean Average Precision), and NDCG (Normalized Discounted Cumulative Gain) (see their definitions in Appendix C.2).
**(5) Algorithms.** We implement all algorithms using Visual C++.

Figure 5: Accuracy for Edge-To-Edge Similarities

| Algo. | Description |
|-------|-------------|
| SE | Iterative algorithm in Theorem 4 to assess all edge-to-edge similarities $r(*, *)$ and all node-to-node similarities $s(*, *)$ |
| SE-PE | Algorithm 1 that evaluates $r(*, e_i)$ *w.r.t.* edge query $e_i$ |
| SE-PN | Algorithm 2 that evaluates $s(*, v_i)$ *w.r.t.* node query $v_i$ |
| SR | SimRank, an influential node-to-node similarity measure [15] |
| SR* | SimRank*, the state-of-the-art variation of SimRank [20] |
| PR | Penetrating-Rank considering both in- and out-links [22] |
| RWR | a popular proximity model – Random Walk with Restart [9] |

**(6) Parameters.** We set (a) $\gamma = 0.6$, as used by SimRank. (b) $k = 20$, which guarantees $\mathbf{R}_k$ and $\mathbf{S}_k$ accurate to $10^{-9}$.

**(7) Ground Truth.** (a) To establish the ground truth for the similarity of neurons (nodes) and synapses (edges) on NU, an evaluation is carried out manually by 22 bioinformatics researchers. The judgment is based on evaluator's knowledge on neuron description, type of synapse, and the affinity view of neuronal topology. Each pair of neurons (nodes) is assigned a relevance score based on neuron name, position of cell body, length of neuron span, and the number of synapses in the head/mid-body/tail. Each pair of synapses (edges) is given a relevance score based on its type, *e.g.* "receives", "sends", "electrical junction" and "neuromuscular junction".

(b) To build the ground truth for similar authors (nodes) and co-authored publications (edges) on DP, we invited 20 researchers from DB, IR, and NC groups. Each pair of authors (nodes) is assigned a score based on their "collaboration distance" in Microsoft Academic Search [3]. Each pair of papers (edges) is given a relevance score based on the assessor's domain-specific knowledge of paper contents, citation relations, and publication venues.

(c) To create the ground truth of relevant airports (nodes) and routes (edges) on AL, we invited 10 travel consultants from Egencia. The judgment relies on the evaluator's experience and knowledge on Airline Route Mapper [1]. Each pair of airports (nodes) is given a relevance score based on their connectivity and closeness; and each pair of routes (edges) is assigned a relevance score based on their airline, the number of overlapped source/destination airports, and the country/territory where the airline is incorporated.

All experiments are run with Intel Xeon E7-8890 v3 processors (18 cores per socket) and 1 TB of memory.

## 7.2 Experimental Results

*7.2.1 Semantic Accuracy.* To evaluate the semantic accuracy of SimEdge, we adopt MAP and NDCG measures on real NU, DP, and AL. For each dataset, we randomly select 300 edge queries $e_q$ to evaluate edge-to-edge similarities $r(*, e_q)$, and 300 node queries $v_q$ to evaluate node-to-node similarities $s(*, v_q)$, respectively.

Fig. 5 shows the average quantitative results on edge-to-edge similarities. We see that (a) SE and SE-PE consistently have high semantic accuracy, *e.g.*, on DP, the average MAP@100 and NDCG@100 for both SE and SE-PE are 0.97 and 0.95, respectively. This highlights the good quality of SimEdge in quantifying edge-to-edge similarity.

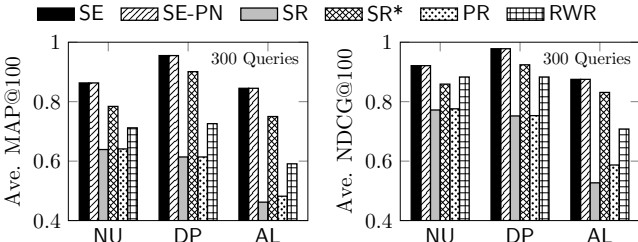

Figure 6: Accuracy for Node-To-Node Similarities

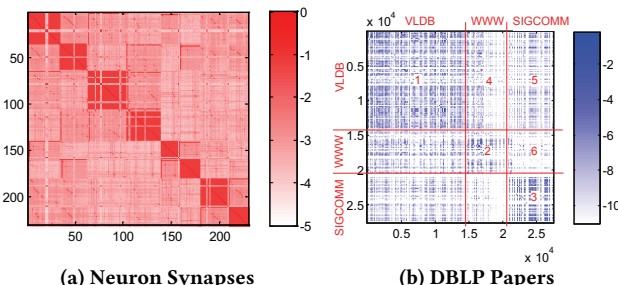

**(a) Neuron Synapses**      **(b) DBLP Papers**

Figure 7: Two Case Studies for Qualitative Analysis

(b) On each dataset, the accuracy of SE is the same as that of SE-PE, showing the correctness of SE-PE by Theorem 9.

Fig. 6 compares the average quantitative results of SE, SE-PN with SR, SR*, PR, RWR for node-to-node similarities. The results show that (a) on every dataset, SE and SE-PN exhibit the highest accuracy for node-to-node evaluation, *e.g.* on NU, the NDCG@100 for both SE and SE-PN is 0.921, much superior to SR (0.772), SR* (0.859), PR (0.776), RWR (0.883). This is because SE and SE-PN are transitive measures that fulfill the triangle inequality so that all weakly connected paths between nodes are captured. In contrast, SR, SR*, PR, RWR often neglect certain weakly connected paths, thereby yielding less meaningful results. (b) The huge discrepancy in accuracy between SE and SR validates that SimEdge can better resolve the "zero-similarity" problems of SimRank than SimRank*.

Fig. 7 depicts the qualitative accuracy of SE for assessing edge-to-edge similarities on NU and DP. For each dataset, we first calculate its SimEdge matrix $\mathbf{R}$, and then rearrange both column- and row-indices of $\mathbf{R}$ according to the ground truth of edges.

Fig. 7a displays the heat map for the first 230 rows and columns of the reordered edge-to-edge SimEdge matrix on NU, in which the darker color of the point in the matrix denotes the higher edge-to-edge SimEdge score. We discern that, after rearrangement, the darker color points mostly are centred on the diagonal blocks of the matrix. This implies that the synapse (edge) communities identified by SE agree well with the ground truth data, showing the superiority of our edge-to-edge SimEdge for edge classification.

Similarly, Fig. 7b visualizes the heat map for all pairs of the reordered edge-to-edge SimEdge matrix on DP. Notice that the darker points are centred on 3 diagonal blocks of the matrix. This suggests that SimEdge effectively identifies 3 communities of papers (edges), corresponding to 3 venues (VLDB, WWW, SIGCOMM). Moreover, the points in Block 5 are darker than those in Blocks 4 and 6. This indicates that the connection between SIGCOMM and VLDB papers is stronger than the correlations between SIGCOMM and WWW papers, as well as VLDB and WWW papers.

*7.2.2 Computational Time & Memory Efficiency.* Fig. 8a evaluates the computational time for all-pairs of edges on NU, DP, and AL. (1)

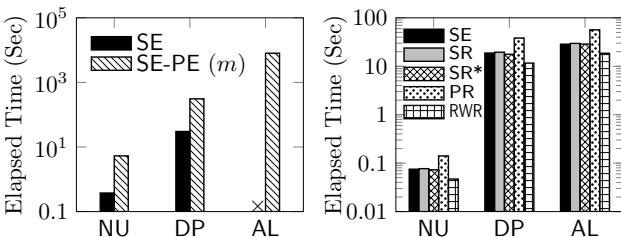

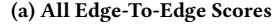

(a) All Edge-To-Edge Scores  (b) All Node-To-Node Scores

**Figure 8: Computational Time on Real Datsets**

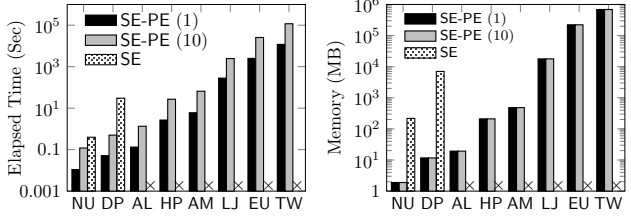

**Figure 9: Time & Memory for Assessing Edge-To-Edge Similarities *w.r.t.* # of Queries on Real Datasets**

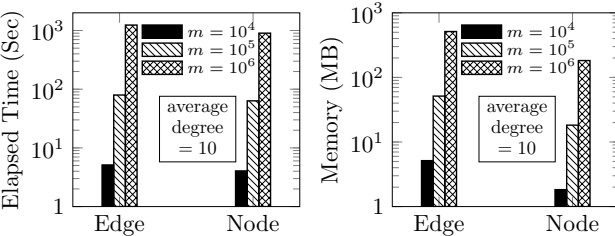

**Figure 10: Time & Memory of SE-PE (*resp.* SE-PN) for All Pairs of Edges (*resp.* Nodes) Evaluation on Synthetic Data**

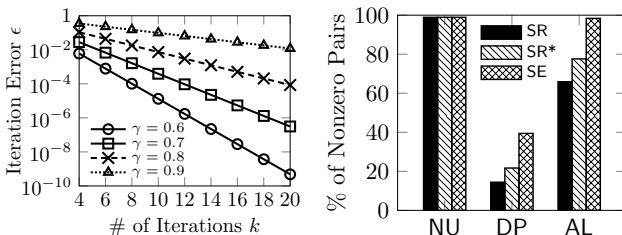

**Figure 11: Convergence Rate**     **Figure 12: # of Nonzero Pairs**

On each dataset, when SE does not fail for edge-to-edge similarity evaluation, SE runs slightly faster than execution of SE-PE $m$ times. This is because SE-PE will break down the computation of the entire edge-to-edge similarity matrix $\mathbf{R}$ into small pieces consisting of $m$ columns, $\mathbf{R}_{\star,e}$, $\forall e \in E$, This process entails repeated preprocessing to obtain $\mathbf{M}$ and $\mathbf{N}$. Thus, for smaller graphs that can hold the entire $\mathbf{R}$ in memory, SE is slightly faster than SE-PE. (2) On AL, SE fails due to insufficient memory to store the entire $\mathbf{R}$. Consequently, for larger graphs, SE-PE exhibits superior scalability.

Fig. 8b compares the time of SE with others for assessing all-pairs node-to-node similarities. The results show that the time for SE, SR, and SR$^\star$ is nearly identical, all of which are faster than PR. This indicates that SE does not need to sacrifice extra time for archiving high accuracy to traverse more paths neglected by SR and SR$^\star$, unlike PR entailing more time to capture out-links.

Fig. 9 compares the time and memory of SE-PE with SE to assess edge-to-edge similarities *w.r.t.* queries on real datasets. For each dataset, when varying the query size, we observe that: (1) With a fixed number of queries, both the time and memory for SE-PE grow *w.r.t.* the size of graphs. This is consistent with SE-PE complexity analysis in Theorem 10. (2) SE survives only on small NU and DP, and requires huge memory to store the entire edge-to-edge similarity matrix for the next iteration, even if only a fraction of columns are needed. (3) The time of SE-PE increases with the size of queries, but its memory retains the same, highlighting its scalability. (4) For large datasets, SE-PE also scales well, whereas SE crashes due to memory explosion.

Regarding node-to-node similarity assessment, the performance of SE-PN is similar to that of SE-PE. The results are shown in Appendix C.3 (Fig. 13). Given similar trends, we omit its description.

To assess the efficiency of SimEdge on synthetic datasets, we vary $m$ from $10^4$ to $10^6$ (*resp. n* from $10^3$ to $10^5$). Fig. 10 depicts the time and memory of SE-PE (*resp.* SE-PN) to assess all pairs of edges (*resp.* nodes). We discern that (1) the time and memory of SE-PE (*resp.* SE-PN) linearly increase with the growing size of $n$. This agrees well with the complexity analysis in Theorem 10. (2) The memory of SE-PE is consistently larger than that of SE-PN. This conforms to our intuition that, for the generated graphs,

$m$ is greater than $n$, which implies that edge-to-edge similarity $\mathbf{R}$ contains more nonzeros than node-to-node similarity $\mathbf{S}$.

*7.2.3 Effect of k and γ on Iterative Error.* Fig. 11 shows the effect of the number of iterations $k$ and the damping factor $\gamma$ on the convergence rate of SimEdge. For every fixed $\gamma$, we vary $k$ from 4 from 20, and evaluate the iterative error $\epsilon = \|\mathbf{R}_k - \mathbf{R}\|_{\max}$. We regard $\mathbf{R}_{100}$ as the ideal $\mathbf{R}$, since $k = 100$ ensures the first 15 decimal places of all entries in $\mathbf{R}_k$ and $\mathbf{R}_{k+1}$ to be the same. We see that (1) the error $\epsilon$ dramatically drops as $k$ grows. As $y$ axis is in log scale, the linear trend implies that the rate of convergence is exponential. (2) When $\gamma$ rises from 0.6 to 0.9, the slope of the declining trendline becomes less steep. This implies that a smaller $\gamma$ results in a quicker rate of convergence, which aligns with the error bound in Theorem 5.

*7.2.4 Statistics on % of Nonzero Similarity Pairs.* To show the number of node pairs that are ignored by SR and SR$^\star$ but can be captured by SE, Fig. 12 shows statistical findings on the percentage of nonzero pairs on real datasets for each algorithm. (1) On DP and AL, SE captures far more pairs of nodes with nonzero similarities than SR and SR$^\star$, highlighting its more meaningful semantics. (2) On NU, all the methods can capture ~98.9% nonzero pairs, due to the fewer number of cycles in neural networks. Hence, SR and SR$^\star$ do not always yield unreasonable scores, depending on graph structures.

## 8  CONCLUSION

In this paper, we introduce SimEdge, a novel similarity measure to assess edge-to-edge similarity while preserving transitivity. The core idea behind SimEdge is a mutual reinforcement co-recursion. We show that SimEdge ensures the transitivity of similarity, improving the accuracy of node-to-node SimRank similarity without misrepresenting dissimilarity between nodes. For large-scale graphs, we also present efficient techniques to compute SimEdge similarities using linear memory without any loss of accuracy. Our empirical evaluation on various datasets validates 1) the high accuracy of SimEdge in capturing transitive edge-to-edge relationships, 2) a more reliable assessment of node-to-node similarity than SimRank and its variants, and 3) the superior scalability and fast speed on large graphs with billions of edges.

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

# A  PROOF OF THEOREMS & LEMMAS

## A.1  Proof of Theorem 1

PROOF. By Definition 2, the four components (1a)–(1d) in Eq.(1) can be rewritten, respectively, as

$$s(H(e_a),H(e_b)) = [\mathbf{B}^T\mathbf{S}\mathbf{B}]_{e_a,e_b} \quad s(H(e_a),T(e_b)) = [\mathbf{B}^T\mathbf{S}\mathbf{A}]_{e_a,e_b}$$

$$s(T(e_a),H(e_b)) = [\mathbf{A}^T\mathbf{S}\mathbf{B}]_{e_a,e_b} \quad s(T(e_a),T(e_b)) = [\mathbf{A}^T\mathbf{S}\mathbf{A}]_{e_a,e_b}$$

Substituting the above results into Eq.(1) yields

$$[\mathbf{R}]_{e_a,e_b} = \tfrac{\gamma}{4}[\mathbf{B}^T\mathbf{S}\mathbf{B} + \mathbf{B}^T\mathbf{S}\mathbf{A} + \mathbf{A}^T\mathbf{S}\mathbf{B} + \mathbf{A}^T\mathbf{S}\mathbf{A}]_{e_a,e_b} + (1-\gamma)[\mathbf{I}_m]_{e_a,e_b}$$

$$= \gamma \cdot \left[\left(\tfrac{\mathbf{B}+\mathbf{A}}{2}\right)^T \mathbf{S}\left(\tfrac{\mathbf{B}+\mathbf{A}}{2}\right)\right]_{e_a,e_b} + (1-\gamma)\cdot[\mathbf{I}_m]_{e_a,e_b}$$

Hence, combining $\mathbf{N} = \tfrac{1}{2}(\mathbf{A}+\mathbf{B})$, we can obtain Eq.(3a).

Next, let us prove Eq.(3b). By the definition of $\bar{\mathbf{A}}$ and $\bar{\mathbf{B}}$, the first component (2a) of Eq.(2) can be expressed as

$$(2a) = \frac{\sum_{v_x\in L^-(v_a)}\sum_{v_y\in L^-(v_b)}\mathbf{R}_{v_x,v_y}}{|L^-(v_a)||L^-(v_b)|}$$

$$= \sum_{v_x\in V}\bar{\mathbf{B}}_{v_a,e_x}\sum_{v_y\in V}\bar{\mathbf{B}}_{v_b,e_y}\mathbf{R}_{e_x,e_y} = \left[\bar{\mathbf{B}}\mathbf{R}\bar{\mathbf{B}}^T\right]_{v_a,v_b}$$

Similarly, the components (2b)–(2d) can be represented as

$$(2b) = \left[\bar{\mathbf{B}}\mathbf{R}\bar{\mathbf{A}}^T\right]_{v_a,v_b} \quad (2c) = \left[\bar{\mathbf{A}}\mathbf{R}\bar{\mathbf{A}}^T\right]_{v_a,v_b} \quad (2d) = \left[\bar{\mathbf{A}}\mathbf{R}\bar{\mathbf{A}}^T\right]_{v_a,v_b}$$

Plugging the above results into Eq.(2) produces

$$[\mathbf{S}]_{v_a,v_b} = \tfrac{\gamma}{4}\left[\bar{\mathbf{B}}\mathbf{R}\bar{\mathbf{B}}^T + \bar{\mathbf{B}}\mathbf{R}\bar{\mathbf{A}}^T + \bar{\mathbf{A}}\mathbf{R}\bar{\mathbf{B}}^T + \bar{\mathbf{A}}\mathbf{R}\bar{\mathbf{A}}^T\right]_{v_a,v_b} + (1-\gamma)[\mathbf{I}_n]_{v_a,v_b}$$

$$= \gamma\cdot\left[\left(\tfrac{\bar{\mathbf{B}}+\bar{\mathbf{A}}}{2}\right)\mathbf{R}\left(\tfrac{\bar{\mathbf{B}}+\bar{\mathbf{A}}}{2}\right)^T\right]_{v_a,v_b} + (1-\gamma)\cdot[\mathbf{I}_n]_{v_a,v_b}$$

Thus, utilizing $\mathbf{M} = \tfrac{1}{2}(\bar{\mathbf{A}}+\bar{\mathbf{B}})$, we have Eq.(3b). □

## A.2  Proof of Theorem 2

PROOF. We focus on showing the existence and uniqueness of $\mathbf{R}$. A similar proof can be applied to $\mathbf{S}$ and is omitted for brevity.

Substituting $\mathbf{S}$ of Eq.(3b) back into Eq.(3a) yields

$$\mathbf{R} = \gamma^2(\mathbf{N}^T\mathbf{M})\mathbf{R}(\mathbf{N}^T\mathbf{M})^T + (1-\gamma)(\gamma\mathbf{N}^T\mathbf{N} + \mathbf{I}_m) \quad (10)$$

Let $\mathcal{L}(*)$ be a linear transformation from $\mathbb{R}^{m\times m}$ to $\mathbb{R}^{m\times m}$:

$$\mathcal{L}(\mathbf{X}) \triangleq \mathbf{X} - \gamma^2(\mathbf{N}^T\mathbf{M})\mathbf{X}(\mathbf{N}^T\mathbf{M})^T \quad (\text{for } \mathbf{X}\in\mathbb{R}^{m\times m})$$

Then, Eq.(10) is expressible as $\mathcal{L}(\mathbf{R}) = (1-\gamma)(\gamma\mathbf{N}^T\mathbf{N}+\mathbf{I}_m)$. Thus, to prove the existence and uniqueness of $\mathbf{R}$, we just need to show that $\mathcal{L}(*)$ is invertible, or equivalently, $\mathcal{L}(*)$ has no 0 eigenvalue.

Let $\mathbf{u}_1,\cdots,\mathbf{u}_\tau$ be the eigenvectors of the matrix $(\mathbf{N}^T\mathbf{M})$ associated with the eigenvalues $\rho_1,\cdots,\rho_\tau$, i.e.,

$$(\mathbf{N}^T\mathbf{M})\mathbf{u}_i = \rho_i\mathbf{u}_i \quad (\forall i=1,2,\cdots,\tau)$$

Then, it follows that, $\forall i=1,2,\cdots,\tau$ and $\forall j=1,2,\cdots,\tau$

$$\mathcal{L}(\mathbf{u}_i\mathbf{u}_j^T) = \mathbf{u}_i\mathbf{u}_j^T - \gamma^2(\mathbf{N}^T\mathbf{M})\mathbf{u}_i\mathbf{u}_j^T(\mathbf{N}^T\mathbf{M})^T = (1-\gamma^2\rho_i\rho_j)\mathbf{u}_i\mathbf{u}_j^T$$

meaning that $\mathcal{L}(*)$ has $\tau^2$ eigenvalues $(1-\gamma^2\rho_i\rho_j)$ $(\forall i,j)$.

Next, we will show that all eigenvalues of $\mathcal{L}(*)$ are not 0s. Due to $\gamma\in(0,1)$, $1-\gamma^2\rho_i\rho_j \neq 0$ $(\forall i,j) \Leftrightarrow \max_i\{|\rho_i|\}\leq 1$.

By $\mathbf{N}$ and $\mathbf{M}$ definition, for each row of $(\mathbf{N}^T\mathbf{M})$, we have

$$[\mathbf{N}^T\mathbf{M}]_{x,*} = \tfrac{1}{4}[\mathbf{A}^T+\mathbf{B}^T]_{x,*}(\bar{\mathbf{A}}+\bar{\mathbf{B}}) \quad (\forall x=1,\cdots,m)$$

Since each edge is connected by two nodes (head and tail), there are only two 1s entries in every row of $(\mathbf{A}^T+\mathbf{B}^T)$. Let $y_1$ and $y_2$ be the column indices of the two 1s entries in $[\mathbf{A}^T+\mathbf{B}^T]_{x,*}$. Since other entries of row $x$ are 0s, we have

$$[\mathbf{N}^T\mathbf{M}]_{x,*} = \tfrac{1}{4}([\bar{\mathbf{A}}+\bar{\mathbf{B}}]_{y_1,*} + [\bar{\mathbf{A}}+\bar{\mathbf{B}}]_{y_2,*})$$

Taking $\|*\|_\infty$ norms on both sides, we can get

$$\left\|[\mathbf{N}^T\mathbf{M}]_{x,*}\right\|_\infty = \tfrac{1}{4}\left\|[\bar{\mathbf{A}}+\bar{\mathbf{B}}]_{y_1,*} + [\bar{\mathbf{A}}+\bar{\mathbf{B}}]_{y_2,*}\right\|_\infty$$

$$= \tfrac{1}{4}((1+1)+(1+1)) \leq 1 \quad (\forall x=1,\cdots,m)$$

By spectral radius property, $\max_i\{|\rho_i|\} \leq \|\mathbf{N}^T\mathbf{M}\|_\infty \leq 1$. □

## A.3  Proof of Theorem 3

PROOF. We just show Eq.(4), and a similar proof applies to Eq.(5). Let $\mathbf{Q} = \gamma(\mathbf{N}^T\mathbf{M})$ and $\mathbf{C} = (1-\gamma)(\gamma\mathbf{N}^T\mathbf{N}+\mathbf{I}_m)$. Then, Eq.(10) in Theorem 2 can be rewritten as

$$\mathbf{R} = \mathbf{Q}\mathbf{R}\mathbf{Q}^T + \mathbf{C} \quad \Leftrightarrow \quad \mathbf{R} = \sum_{k=0}^{\infty}\mathbf{Q}^k\mathbf{C}(\mathbf{Q}^T)^k$$

In virtue of Theorem 2, the above series of $\mathbf{R}$ is convergent since $\|\mathbf{Q}\|_\infty \leq \gamma < 1$. Plugging the definition of $\mathbf{Q}$ and $\mathbf{C}$ back into the above series produces Eq.(4). □

## A.4 Proof of Theorem 4

PROOF. Successive substitution applied to Eq.(6) yields

$$\mathbf{R}_l = (1-\gamma) \sum_{k=0}^{l} \gamma^{2k} (\mathbf{N}^T\mathbf{M})^k (\gamma\mathbf{N}^T\mathbf{N} + \mathbf{I}_m)(\mathbf{M}^T\mathbf{N})^k \quad (11)$$

Thus, (a) holds. To prove (b), we plug Eq.(11) into Eq.(6c)

$$\mathbf{S}_l = (1-\gamma)\left( \sum_{k=0}^{l-1} \gamma^{2k+1} \mathbf{M}(\mathbf{N}^T\mathbf{M})^k (\gamma\mathbf{N}^T\mathbf{N} + \mathbf{I}_m)(\mathbf{M}^T\mathbf{N})^k \mathbf{M}^T + \mathbf{I}_n \right)$$

$$= (1-\gamma)\left( \sum_{k=0}^{l-1} \gamma^{2k+2} \overbrace{\mathbf{M}(\mathbf{N}^T\mathbf{M})^k}^{=(\mathbf{M}\mathbf{N}^T)^{k+1}} (\mathbf{N}^T\mathbf{N}) \overbrace{(\mathbf{M}^T\mathbf{N})^k \mathbf{M}^T}^{=(\mathbf{N}\mathbf{M}^T)^{k+1}} \right.$$

$$\left. + \sum_{k=0}^{l-1} \gamma^{2k+1} \underbrace{\mathbf{M}(\mathbf{N}^T\mathbf{M})^k (\mathbf{M}^T\mathbf{N})^k \mathbf{M}^T}_{=(\mathbf{M}\mathbf{N}^T)^k (\mathbf{M}\mathbf{M}^T)(\mathbf{N}\mathbf{M}^T)^k} + \mathbf{I}_n \right)$$

$$= (1-\gamma)\left( \overbrace{\sum_{k=1}^{l} \gamma^{2k} (\mathbf{M}\mathbf{N}^T)^k (\mathbf{N}\mathbf{M}^T)^k}^{=\gamma^{2l}(\mathbf{M}\mathbf{N}^T)^l(\mathbf{N}\mathbf{M}^T)^l + \sum_{k=1}^{l-1}(\ldots)} \right.$$

$$\left. + \underbrace{\sum_{k=0}^{l-1} \gamma^{2k} (\mathbf{M}\mathbf{N}^T)^k (\gamma\mathbf{M}\mathbf{M}^T)(\mathbf{N}\mathbf{M}^T)^k + \mathbf{I}_n}_{=\gamma\mathbf{M}\mathbf{M}^T + \sum_{k=1}^{l-1}(\ldots)} \right)$$

$$= (1-\gamma)\gamma^{2l} (\mathbf{M}\mathbf{N}^T)^l (\mathbf{N}\mathbf{M}^T)^l$$
$$+ \{\text{the first } (l-1)\text{-th partial sums of Eq.(5)}\} \quad (12)$$

Hence, (b) holds. To prove (c), taking the limit on both sides of Eq.(11) yields Eq.(4). Thus, $\lim_{l\to\infty} \mathbf{R}_l = \mathbf{R}$.

We can verify that $\|\mathbf{M}\mathbf{N}^T\|_\infty \leq 1$, which implies that

$$\|(1-\gamma)\gamma^{2l}(\mathbf{M}\mathbf{N}^T)^l(\mathbf{N}\mathbf{M}^T)^l\|_\infty \leq (1-\gamma)\gamma^{2l} \to 0 \ (l \to \infty)$$

By taking the limit on both sides of Eq.(12) yields Eq.(5). □

## A.5 Proof of Theorem 5

PROOF. We can readily verify that, for two vectors $\mathbf{x}$ and $\mathbf{y} \in \mathbb{R}^{m\times 1}$ and a matrix $\mathbf{Z} \in \mathbb{R}^{m\times m}$, if $\|\mathbf{x}\|_\infty = \|\mathbf{y}\|_\infty = 1$,

$$\mathbf{x}\mathbf{Z}\mathbf{y}^T \leq (\mathbf{x}\mathbf{1}^T\mathbf{y}^T)\|\mathbf{Z}\|_{\max} = \|\mathbf{Z}\|_{\max} \text{ with } \mathbf{1} = (1,\cdots,1)^T$$

By Theorems 3 and 4, we subtract Eq.(11) from Eq.(4):

$$[\mathbf{R} - \mathbf{R}_l]_{i,j} = (1-\gamma) \sum_{k=l+1}^{\infty} \gamma^{2k} \underbrace{[(\mathbf{N}^T\mathbf{M})^k]_{i,*}}_{=\mathbf{x}} \underbrace{(\gamma\mathbf{N}^T\mathbf{N} + \mathbf{I}_m)}_{=\mathbf{Z}} \underbrace{[(\mathbf{M}^T\mathbf{N})^k]_{*,j}}_{=\mathbf{y}^T}$$

Since $\|\mathbf{N}^T\mathbf{M}\|_\infty \leq 1$ and $\|\mathbf{N}^T\mathbf{N}\|_{\max} \leq 1$, it follows that

$$\|\mathbf{R} - \mathbf{R}_l\|_{\max} \leq (1-\gamma)\sum_{k=l+1}^{\infty} \gamma^{2k}(\gamma\|\mathbf{N}^T\mathbf{N}\|_{\max} + 1) = \gamma^{2(l+1)}$$

Similarly, utilizing $\|\mathbf{M}\mathbf{N}^T\|_\infty \leq 1$ and $\|\mathbf{M}^T\mathbf{M}\|_{\max} \leq 1$, we can prove that $\|\mathbf{S} - \mathbf{S}_l\|_{\max} \leq \gamma^{2l}$. □

## A.6 Proof of Lemma 1

PROOF. By the definition of $\mathbf{S}$ in Eq.(2) and its iterative form in Eq.(6c), for any three nodes $v_a, v_b, v_c$, it follows that

$$[\mathbf{S}_{k+1}]_{v_a,v_c} - [\mathbf{S}_{k+1}]_{v_a,v_b} = \frac{\gamma}{4} \times \Big($$

$$\underset{(e_i,e_j)\in L^-(v_a)\times L^-(v_c)}{\text{average}} \{[\mathbf{R}_k]_{e_i,e_j}\} - \underset{(e_i,e_j)\in L^-(v_a)\times L^+(v_b)}{\text{average}} \{[\mathbf{R}_k]_{e_i,e_j}\} \quad (13a)$$

$$+ \underset{(e_i,e_j)\in L^-(v_a)\times L^+(v_c)}{\text{average}} \{[\mathbf{R}_k]_{e_i,e_j}\} - \underset{(e_i,e_j)\in L^-(v_a)\times L^-(v_b)}{\text{average}} \{[\mathbf{R}_k]_{e_i,e_j}\} \quad (13b)$$

$$+ \underset{(e_i,e_j)\in L^+(v_a)\times L^-(v_c)}{\text{average}} \{[\mathbf{R}_k]_{e_i,e_j}\} - \underset{(e_i,e_j)\in L^+(v_a)\times L^-(v_b)}{\text{average}} \{[\mathbf{R}_k]_{e_i,e_j}\} \quad (13c)$$

$$+ \underset{(e_i,e_j)\in L^+(v_a)\times L^+(v_c)}{\text{average}} \{[\mathbf{R}_k]_{e_i,e_j}\} - \underset{(e_i,e_j)\in L^+(v_a)\times L^+(v_b)}{\text{average}} \{[\mathbf{R}_k]_{e_i,e_j}\}\Big) \quad (13d)$$

$$+ \begin{cases} 1-\gamma & (v_a = v_c) \\ 0 & (v_a \neq v_c) \end{cases} - \begin{cases} 1-\gamma & (v_a = v_b) \\ 0 & (v_a \neq v_b) \end{cases} \quad (13e)$$

Let us first evaluate the first part (13a). By the definition of average$(*,*)$, it follows that

$$(13a) = \frac{\sum_{e_i\in L^-(v_a)}\sum_{e_j\in L^-(v_c)}[\mathbf{R}_k]_{e_i,e_j}}{|L^-(v_a)||L^-(v_c)|} - \frac{\sum_{e_i\in L^-(v_a)}\sum_{e_j\in L^+(v_b)}[\mathbf{R}_k]_{e_i,e_j}}{|L^-(v_a)||L^+(v_b)|}$$

$$= \frac{1}{|L^-(v_a)|}\sum_{e_i\in L^-(v_a)}\phi(e_i), \quad (14)$$

$$\text{where} \quad \phi(e_i) \triangleq \frac{1}{|L^-(v_c)|}\sum_{e_j\in L^-(v_c)}[\mathbf{R}_k]_{e_i,e_j}$$
$$- \frac{1}{|L^+(v_b)|}\sum_{e_j\in L^+(v_b)}[\mathbf{R}_k]_{e_i,e_j}.$$

To find out the lower bound for $\phi(e_i)$, we notice that

$$\sum_{e_j\in L^-(v_c)}[\mathbf{R}_k]_{e_i,e_j} = \frac{1}{|L^+(v_b)|}\sum_{e_y\in L^+(v_b)}\sum_{e_x\in L^-(v_c)}[\mathbf{R}_k]_{e_i,e_x}$$

$$\sum_{e_j\in L^+(v_b)}[\mathbf{R}_k]_{e_i,e_j} = \frac{1}{|L^-(v_c)|}\sum_{e_y\in L^+(v_b)}\sum_{e_x\in L^-(v_c)}[\mathbf{R}_k]_{e_i,e_y},$$

which implies that

$$\phi(e_i) = \frac{\sum_{e_y\in L^+(v_b)}\sum_{e_x\in L^-(v_c)}\left([\mathbf{R}_k]_{e_i,e_x}-[\mathbf{R}_k]_{e_i,e_y}\right)}{|L^-(v_c)||L^+(v_b)|}$$

$$\geq \frac{\sum_{e_y\in L^+(v_b)}\sum_{e_x\in L^-(v_c)}\left([\mathbf{R}_k]_{e_y,e_x}-1\right)}{|L^-(v_c)||L^+(v_b)|}$$

$$= \underset{(e_y,e_x)\in L^-(v_b)\times L^+(v_c)}{\text{average}} \{[\mathbf{R}_k]_{e_y,e_x}\} - 1$$

Thus, substituting the above lower bound into Eq.(14) yields

$$(13a) \geq \underset{(e_y,e_x)\in L^-(v_b)\times L^+(v_c)}{\text{average}} \{[\mathbf{R}_k]_{e_y,e_x}\} - 1$$

Similarly, we can prove that

$$(13b) \geq \underset{(e_y,e_x)\in L^+(v_b)\times L^-(v_c)}{\text{average}} \{[\mathbf{R}_k]_{e_y,e_x}\} - 1$$

$$(13c) \geq \underset{(e_y,e_x)\in L^-(v_b)\times L^-(v_c)}{\text{average}} \{[\mathbf{R}_k]_{e_y,e_x}\} - 1$$

$$(13d) \geq \underset{(e_y,e_x)\in L^+(v_b)\times L^+(v_c)}{\text{average}} \{[\mathbf{R}_k]_{e_y,e_x}\} - 1$$

We next evaluate the last part (13e) as follows:

$$(13e) = \begin{cases} 1-\gamma & (\text{if } v_a = v_c \text{ and } v_a \neq v_b) \\ \gamma - 1 & (\text{if } v_a \neq v_c \text{ and } v_a = v_b) \\ 0 & (\text{otherwise}) \end{cases}$$

$$\geq \begin{cases} \min\{1-\gamma, \gamma-1, 0\} = \gamma - 1 & (v_b \neq v_c) \\ 0 & (v_b = v_c) \end{cases}$$

Finally, we replace (13a)–(13e) with the above lower bounds:

$$[S_{k+1}]_{v_a,v_c} - [S_{k+1}]_{v_a,v_b} \geq \frac{\gamma}{4} \times \Big( \operatorname*{average}_{(e_y,e_x) \in L^-(v_b) \times L^+(v_c)} \{[R_k]_{e_i,e_j}\} - 1$$
$$+ \operatorname*{average}_{(e_y,e_x) \in L^+(v_b) \times L^-(v_c)} \{[R_k]_{e_i,e_j}\} - 1$$
$$+ \operatorname*{average}_{(e_y,e_x) \in L^-(v_b) \times L^-(v_c)} \{[R_k]_{e_i,e_j}\} - 1$$
$$+ \operatorname*{average}_{(e_y,e_x) \in L^+(v_b) \times L^+(v_c)} \{[R_k]_{e_i,e_j}\} - 1 \Big)$$
$$+ \begin{cases} \gamma - 1 & (v_b \neq v_c) \\ 0 & (v_b = v_c) \end{cases}$$
$$= [S_{k+1}]_{v_b,v_c} - 1$$

□

## A.7 Proof of Lemma 2

PROOF. By the definition of $R$ in Eq.(1) and its iterative form in Eq.(6b), for any three edges $e_a, e_b, e_c$, it follows that

$$[R_k]_{e_a,e_c} - [R_k]_{e_a,e_b} = \frac{\gamma}{4} \times \Big( [S_k]_{H(e_a),H(e_c)} - [S_k]_{H(e_a),H(e_b)}$$
$$+ [S_k]_{H(e_a),T(e_c)} - [S_k]_{H(e_a),T(e_b)}$$
$$+ [S_k]_{T(e_a),H(e_c)} - [S_k]_{T(e_a),H(e_b)}$$
$$+ [S_k]_{T(e_a),T(e_c)} - [S_k]_{T(e_a),T(e_b)} \Big)$$
$$+ \begin{cases} 1 - \gamma & (\text{if } e_a = e_c \text{ and } e_a \neq e_b) \\ \gamma - 1 & (\text{if } e_a \neq e_c \text{ and } e_a = e_b) \\ 0 & (\text{otherwise}) \end{cases}$$
$$\geq \frac{\gamma}{4} \Big( [S_k]_{H(e_b),H(e_c)} - 1 + [S_k]_{T(e_b),T(e_c)} - 1$$
$$+ [S_k]_{H(e_b),H(e_c)} - 1 + [S_k]_{T(e_b),T(e_c)} - 1 \Big) + \begin{cases} \gamma - 1 & (e_c \neq e_b) \\ 0 & (e_c = e_b) \end{cases}$$
$$= [R_k]_{e_b,e_c} - 1. \qquad (\forall e_a, e_b, e_c \in E)$$

□

## A.8 Proof of Theorem 6

PROOF. Clearly, $0 \leq s(v_a, v_b) \leq 1$ and $s(v_a, v_b) = s(v_b, v_a)$. Thus, (a) and (b) hold.

To prove (c), combining Lemmas 1 and 2, we can get that, for each $k = 0, 1, 2, \cdots$ and any three nodes $v_a, v_b, v_c \in V$, if $S_k$ in Eq.(6b) satisfies

$$[S_k]_{v_a,v_c} - [S_k]_{v_a,v_b} \geq [S_k]_{v_b,v_c} - 1,$$

then $S_{k+1}$ in Eq.(6c) satisfies

$$[S_{k+1}]_{v_a,v_c} - [S_{k+1}]_{v_a,v_b} \geq [S_{k+1}]_{v_b,v_c} - 1.$$

Since the base case of $k = 0$ holds as follows:

$$[S_0]_{v_a,v_c} - [S_0]_{v_a,v_b}$$
$$= \begin{cases} 1-\gamma & (v_a = v_c) \\ 0 & (v_a \neq v_c) \end{cases} - \begin{cases} 1-\gamma & (v_a = v_b) \\ 0 & (v_a \neq v_b) \end{cases}$$
$$\geq \begin{cases} \min\{1-\gamma, \gamma-1, 0\} = \gamma-1 & (v_b \neq v_c) \\ 0 & (v_b = v_c) \end{cases}$$
$$\geq -1 + \begin{cases} 0 & (v_b \neq v_c) \\ 1-\gamma & (v_b = v_c) \end{cases} = [S_0]_{v_b,v_c} - 1,$$

we have $[S_k]_{v_a,v_c} - [S_k]_{v_a,v_b} \geq [S_k]_{v_b,v_c} - 1$ holds for all $k$. Next, taking the limit as $k \to \infty$ gives

$$[S]_{v_a,v_c} - [S]_{v_a,v_b} \geq [S]_{v_b,v_c} - 1$$
$$\Leftrightarrow \quad (1 - [S]_{v_a,v_b}) + (1 - [S]_{v_b,v_c}) \geq (1 - [S]_{v_a,v_c})$$
$$\Leftrightarrow \quad dist_s(v_a, v_b) + dist_s(v_b, v_c) \geq dist_s(v_a, v_c).$$

□

## A.9 Proof of Theorem 7

PROOF. We show that, for any two nodes $v_a$ and $v_b$, if the above constraints are not satisfied, SimRank* $s(v_a, v_b) = 0$. Let us revisit the matrix form of SimRank* in [20]:

$$[S]_{v_a,v_b} = (1-\gamma) \sum_{k=0}^{\infty} \frac{\gamma^k}{2^k} \sum_{\alpha=0}^{k} \binom{k}{\alpha} [(W^T)^\alpha]_{v_a,\star} \cdot [W^{k-\alpha}]_{\star,v_b}$$

For each fixed $k$ and $\alpha \in [0, k]$, if there exists $\alpha_0 \in [0, \alpha-1]$ s.t. $e_{\alpha_0}$ in Eq.(7) bears "$\to$" direction, then by the power property of an adjacency matrix, we have $[(W^T)^\alpha]_{v_a,\star} = 0$, which implies that $[S]_{v_a,v_b} = 0$.

Similarly, for each fixed $k$ and $\alpha \in [0, k]$, if $\exists \alpha_0 \in [\alpha, k-1]$ s.t. $e_{\alpha_0}$ in Eq.(7) bears "$\leftarrow$" direction, then $[W^\alpha]_{\star,v_b} = 0$, implying that $[S]_{v_a,v_b} = 0$. □

## A.10 Proof of Theorem 8

PROOF. We assume that there exists a (weakly connected) path between nodes $v_a$ and $v_b$ that is neglected by SimEdge:

$$(v_a =)v_0, e_0, v_1, e_1, v_2, \cdots, v_{k-1}, e_{k-1}, v_k(= v_b) \qquad (15)$$

and show that this leads to a contradiction. Suppose node-to-node similarity $s(v_0, v_k) = 0$, we can infer from Eq.(2) that its related edge-to-edge similarity $r(e_0, e_{k-1}) = 0$, and recursively, $s(v_1, v_{k-1}) = 0$, $r(e_1, e_{k-2}) = 0$, and so forth. Eventually,
a) if $k$ is an even number, we can arrive at $s(v_{k/2}, v_{k/2}) = 0$, which contradicts that "each node is most similar to itself".
b) if $k$ is an odd number, we have $r(e_{(k-1)/2}, e_{(k-1)/2}) = 0$, contradicting that "each edge is most similar to itself". □

## A.11 Proof of Theorem 9

PROOF. Let $z_l = (\gamma N^T N + I_m) \xi_{k-l}$ and $C = \gamma N^T M$. Then, $\eta_k$ can be iteratively computed from Eq.(8) as

$$\eta_k = C\eta_{k-1} + z_k = C^2 \eta_{k-2} + Cz_{k-1} + z_k$$
$$= C^3 \eta_{k-3} + C^2 z_{k-2} + Cz_{k-1} + z_k$$
$$= \cdots = C^k \eta_0 + C^{k-1} z_1 + \cdots + Cz_{k-1} + z_k \qquad (16)$$

To evaluate $z_1, \cdots, z_k$, we can obtain $\xi_k$ from Eq.(9):

$$\xi_k = (\gamma M^T N) \xi_{k-1} = (\gamma M^T N)^2 \xi_{k-2} = \cdots = (\gamma M^T N)^k \xi_0$$

Thus, it follows that

$$\eta_0 = (I_m + \gamma N^T N) \xi_k = (I_m + \gamma N^T N)(\gamma M^T N)^k \xi_0$$
$$z_l = (\gamma N^T N + I_m) \xi_{k-l} = (\gamma N^T N + I_m)(\gamma M^T N)^{k-l} \xi_0 \quad (\forall l)$$

Substituting $\eta_0$ and $z_l$ and $C = \gamma N^T M$ into Eq.(16) yields

$$\eta_k = \overbrace{(\gamma N^T M)^k}^{C^k =} \overbrace{(I_m + \gamma N^T N)(\gamma M^T N)^k \xi_0}^{\eta_0 =}$$

$$+ \sum_{l=0}^{k-1} \underbrace{(\gamma \mathbf{N}^T \mathbf{M})^l}_{\mathbf{C}^l=} \underbrace{(\gamma \mathbf{N}^T \mathbf{N} + \mathbf{I}_m)(\gamma \mathbf{M}^T \mathbf{N})^l \boldsymbol{\xi}_0}_{\mathbf{z}_{k-l}=}$$

$$= \sum_{l=0}^{k} (\gamma \mathbf{N}^T \mathbf{M})^l (\gamma \mathbf{N}^T \mathbf{N} + \mathbf{I}_m)(\gamma \mathbf{M}^T \mathbf{N})^l [\mathbf{I}_m]_{\star, e_i}$$

Using $[\mathbf{R}_k]_{\star, e_i} = (1 - \gamma)\boldsymbol{\eta}_k$ produces (a) in Theorem 4. □

### A.12 Proof of Theorem 10

Proof. The computational cost consists of two phases:

(a) computing $\boldsymbol{\xi}_0, \cdots, \boldsymbol{\xi}_k$ in Eq.(9). For each iteration $l$, the computational time of Eq.(9) is $O(m)$, dominated by $\mathbf{M}^T (\mathbf{N}\boldsymbol{\xi}_{l-1})$. More concretely, as the number of nonzeros in sparse $\mathbf{N}$ is $O(m)$, it requires $O(m)$ time to compute $(\mathbf{N}\boldsymbol{\xi}_{l-1})$ whose result is an $n \times 1$ vector $\mathbf{x}$; since each row of $\mathbf{M}^T$ has at most two nonzeros, it yields $O(n)$ to compute $\mathbf{M}^T \mathbf{x}$. Hence, for $k$ iterations, it entails $O(km)$ time to compute $\boldsymbol{\xi}_0, \cdots, \boldsymbol{\xi}_k$.

(b) computing $\boldsymbol{\eta}_k$ in Eq.(8). For each iteration $l$, the time complexity required to compute $\boldsymbol{\eta}_0$ and $\boldsymbol{\eta}_l$ are bounded by $\mathbf{N}^T (\mathbf{N}\boldsymbol{\xi}_k)$ and $\mathbf{N}^T (\mathbf{N}\boldsymbol{\xi}_{k-l} + \mathbf{M}\boldsymbol{\eta}_{l-1})$, respectively. We can verify that $(\mathbf{N}\boldsymbol{\xi}_k)$ and $(\mathbf{N}\boldsymbol{\xi}_{k-l} + \mathbf{M}\boldsymbol{\eta}_{l-1})$ requires $O(m)$ time, and their results are two $n \times 1$ vectors, denoted as $\mathbf{x}_1$ and $\mathbf{x}_2$. Thereby, it requires $O(n)$ time to compute $\mathbf{N}^T \mathbf{x}_1$ and $\mathbf{N}^T \mathbf{x}_2$ as each row of $\mathbf{N}^T$ has at most two nonzeros. In total, it requires $O(km)$ time to compute Eq.(9) for $k$ iterations.

By taking (a) and (b) together, the total time is $O(km)$.

For memory usage, it requires $O(m)$ to store $\mathbf{M}$ and $\mathbf{N}$. To store $\boldsymbol{\xi}_0, \cdots, \boldsymbol{\xi}_k$, the $O(km)$ memory is required. Other intermediate vectors (e.g. $(\mathbf{N}\boldsymbol{\xi}_{l-1})$, $(\mathbf{N}\boldsymbol{\xi}_k)$, $(\mathbf{N}\boldsymbol{\xi}_{k-l} + \mathbf{M}\boldsymbol{\eta}_{l-1})$) produce $O(n)$ memory. In total, the memory is dominated by $O(km)$ for $k$ iterations. □

## B ALGORITHM FOR ASSESSING SIMEDGE NODE-TO-NODE SIMILARITY

Algorithm 2 shows the pseudocode for efficiently computing node-to-node SimEdge similarities $[\mathbf{S}]_{\star, v_i}$ *w.r.t.* query node $v_i$.

---

**Algorithm 2:** Evaluate Node-To-Node SimEdge Similarities

**Input** : digraph $G = (V, E)$, query node $v_i \in V$, damping factor $\gamma$, desired accuracy $\epsilon$.
**Output**: node-to-node SimEdge similarities $[\mathbf{S}]_{\star, v_i}$ between all nodes in $G$ and query $v_i$.

1-2 the same as Lines 1–2 in Algorithm 1 ;
3   determine the number of iterations $k \leftarrow \lceil \frac{1}{2} \log_\gamma \epsilon \rceil + 1$ ;
4   initialize $\boldsymbol{\xi}_0 \leftarrow [\mathbf{I}_n]_{\star, v_i}$ ;
5   **for** $l \leftarrow 1, 2, \cdots, k$ **do**
6     |   update $\boldsymbol{\xi}_l \leftarrow \gamma \mathbf{N}(\mathbf{M}^T \boldsymbol{\xi}_{l-1})$ ;
7   initialize $\boldsymbol{\eta}_0 \leftarrow \boldsymbol{\xi}_k + \gamma \mathbf{M}(\mathbf{M}^T \boldsymbol{\xi}_k)$ ;
8   **for** $l \leftarrow 1, 2, \cdots, k$ **do**
9     |   update $\boldsymbol{\eta}_l \leftarrow \boldsymbol{\xi}_{k-l} + \gamma \mathbf{M}(\mathbf{M}^T \boldsymbol{\xi}_{k-l} + \mathbf{N}^T \boldsymbol{\eta}_{l-1})$ ;
10  **return** $[\mathbf{S}_k]_{\star, v_i} \leftarrow (1 - \gamma)\boldsymbol{\eta}_k$ ;

---

## C EXPERIMENTS

### C.1 Detailed Description of Real Datasets

We use six real-world datasets. Below are their detailed description:

(a) NU, a neuronal connectivity graph[2], where each node is a neuron labeled by its name, and each edge is a synapse labeled by its type, *e.g.* Sp (Send-poly), Rp (Receive-poly), EJ (Electric junction), NMJ (Neuromuscular junction).

(b) DP, a co-authorship DBLP network[3], where nodes are authors, and edges represent coauthorship labeled by the latest coauthored paper between two authors. The graph is derived from five-year publications (2010–2015) in VLDB, WWW, and SIGCOMM.

(c) AL, a snapshot of the OpenFlights network[4], where a node denotes an airport labeled with its country or territory, and each edge is a route labeled with its airline code.

(d) HP, a citation network from arXiv high energy physics phenomenology[5], where each node is a paper labeled with its title and abstract, and an edge is a reference.

(e) AM, an Amazon co-purchasing graph, where nodes are products, and edges connect the latest commonly co-purchased items.

(f) LJ, a LiveJournal online social network, where each node is a user, and an edge is a recommendation from one user to another.

(g) EU[6], a large-scale network of EU government web hosting infrastructure, where nodes represent websites or servers, and edges capture the hosting and connectivity relationships.

(h) TW, a massive social network on Twitter, where nodes represent Twitter users, and edges denote follower relationships.

### C.2 Accuracy Metrics

To assess similarity effectiveness, two accuracy metrics are used:

(a) MAP (Mean Average Precision):

$$\text{MAP}(Q) = \frac{1}{|Q|} \sum_{j=1}^{|Q|} \frac{1}{u_j} \sum_{p=1}^{u_j} \text{Precision}(R_{jp})$$

where $u_j$ is the number of relevant objects to query $q_j \in Q$, and $R_{jp}$ is the set of ranked retrieval results from the top results until we get to the $p$-th object.

(b) NDCG (Normalized Discounted Cumulative Gain):

$$\text{NDCG}(Q, p) = \frac{1}{|Q|} \sum_{j=1}^{|Q|} Z_{pj} \sum_{i=1}^{p} \frac{2^{R(j,i)} - 1}{\log_2(1+i)}$$

where $Z_{pj}$ is a normalization factor to ensure a perfect ranking's NDCG at $p$ for query $j$ is 1, and $R(j, i)$ is the relevance score that assessors gave to object $i$ for query $j$.

### C.3 Additional Experiments: Time & Memory for Evaluating Node-To-Node Similarities *w.r.t.* # of Queries on Real Datasets

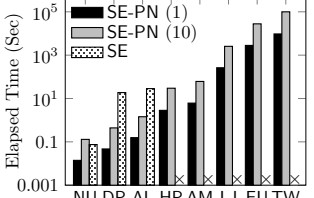 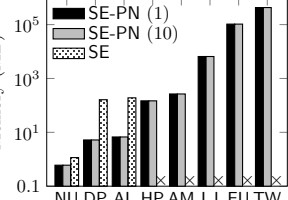

**Figure 13: CPU Time & Memory for Node-To-Node Similarity Assessment *w.r.t.* # of Queries on Real Datasets**

---

[2]http://www.wormatlas.org/neuronalwiring.html
[3]http://dblp.uni-trier.de/˜ley/db/
[4]http://openflights.org/data.html
[5]HP, AM, and LJ are from http://snap.stanford.edu/
[6]EU and TW are taken from LAW (https://law.di.unimi.it/datasets.php)

