# OpenReview forum: "SimEdge: A Scalable Transitivity-Aware Graph-Theoretic Similarity Model for Capturing Edge-to-Edge Relationships"
_ACM.org/TheWebConf/2025/Conference — WWW 2025 Oral_

### Official Review · Reviewer_XxaP · 2024-11-08

**Novelty:** 5
**Technical Quality:** 5

**Review:**

**Summary**

While many exciting similarity measures focus solely on node-to-node similarity, this paper argues that edge-to-edge relationships are equally important. Therefore, we introduce SimEdge, a novel similarity measure designed to assess edge-to-edge similarity while preserving transitivity. The core idea behind SimEdge is a mutual reinforcement co-recursion between node-to-node and edge-to-edge similarity. Thorough theoretical analysis and experimental results have demonstrated the correctness and effectiveness of SimEdge.

**Pros**

1. A novel similarity model that considers edge-to-edge similarity in graphs.
2. The idea of bridging node-to-node and edge-to-edge similarity is both elegant and effective.
3. The detailed theoretical analysis and clear examples make the design section robust and easy to understand.
4. Extensive experiments show that the proposed model outperforms existing methods.

**Cons**

1. **Scope:** SimEdge is more suited for the graph track rather than the search track.

**Other Questions**
1. In Equation (2), should $(v_x, v\_y)$ be $(e\_x, e\_y)$?

**Questions:**

Please refer to the comments above.

**Reviewer Confidence:**

2: The reviewer is willing to defend the evaluation, but it is likely that the reviewer did not understand parts of the paper

**Scope:**

3: The work is somewhat relevant to the Web and to the track, and is of narrow interest to a sub-community

---

### Official Review · Reviewer_RtNz · 2024-11-11

**Novelty:** 4
**Technical Quality:** 6

**Review:**

Quality: In terms of quality, this article provides a comprehensive and detailed explanation of the proposed model, and also uses a large amount of proof and reasoning to demonstrate the effectiveness of the SimEdge model. In addition, the author also discussed how to extend SimEdge in large-scale graphs. Finally, experiments on datasets with multiple characteristics further confirmed SimEdge's high accuracy in capturing node to node (edge to edge) relationships in transmission. The only drawback is that the illustrations in the author's experimental section are almost entirely black and white, which makes it less intuitive for readers to recognize the results of different algorithms. Overall, I believe the quality of the article is high (score 92/100).

Clarity: In terms of clarity, the author first introduced the shortcomings of current research (Simrank, Simrank *) in the introduction section, and based on this, proposed the reasons for the problem, and then introduced SimEdge. Then, the author illustrates the impact of mutual reinforcement and recursion on evaluating node to node (edge to edge) similarity through a simple example, and begins a series of analysis, reasoning, and proof. Finally, the author discussed the extension of the proposed method on large-scale legends and demonstrated the effectiveness of SimEdge through a series of experiments. I think this article has a very clear idea and complete mathematical principles. Therefore, in terms of clarity, I believe it should have a high score (96/100).

Originality: In terms of originality, quantifying the similarity between two objects based on link structure has always been a hot topic. The SimEdge proposed by the author is a new similarity model based on the principle of mutual reinforcement and recursion. I think it is a relatively new method in terms of originality (85/100).

Significance: In terms of research significance, this article aims to propose a new similarity model that can effectively evaluate edge to edge similarity. In terms of application, it can achieve more accurate capture of network filtering, network search, co citation analysis, and other aspects, which is of great significance for the development of the information age. Therefore, the score in terms of meaning is relatively high (85/100).


Advantages of the article:
1. A new similarity model SimEdge has been proposed, which is based on the simple and intuitive principle of mutual reinforcement and recursion, and can effectively evaluate edge to edge similarity.
2. SimEdge is represented in matrix form and a fixed-point iterative method is proposed to simultaneously calculate the edge to edge similarity and node to node similarity of SimEdge.
3. Discussed the extension of the model on large graphs.
4. The proposed model has undergone rigorous reasoning and proof, demonstrating rigor.
5. Extensive experiments have been conducted on authoritative datasets to validate the effectiveness of SimEdge.


Shortcomings of the article:
1. The citation method for some of the work in the Introduction section is not appropriate. For example, in the text "node to node similarity [4, 5, 13, 14, 18, 20, 22]", you can choose the most representative works (no more than 3);
2. The layout of some formulas in the Proposed Model section is not very aesthetically pleasing. For example, the lack of line breaks between definitions (a), (b), and (c) affects the reader's reading experience;
3. The article did not discuss the sensitivity of damping factors to equations 1 and 2 (and thus to SimEdge), but rather described them as empirical values ranging from 0.6 to 0.8. Additional explanations can be provided through analysis or reasoning.
4. The article did not analyze the time complexity of SimEdge. It is recommended to add relevant analysis or experiments.
5. The color scheme in the experimental section is too monotonous. In Figures 5 and 6, different algorithms can be distinguished using different colors to make the experimental results more intuitive.

**Questions:**

1. Can the proposed new metrics be used in some machine learning algorithms to demonstrate the value of SimEdge by controlling the similarity metrics to be different while keeping the machine learning algorithms the same (e.g. K-mean, SVM, etc.).
2. How much impact of damping coefficient on SimEdge, and are there any data preprocessing methods to roughly determine the range of damping coefficient.

**Reviewer Confidence:**

3: The reviewer is confident but not certain that the evaluation is correct

**Scope:**

3: The work is somewhat relevant to the Web and to the track, and is of narrow interest to a sub-community

---

### Official Review · Reviewer_hPp5 · 2024-11-29

**Novelty:** 5
**Technical Quality:** 6

**Review:**

This paper is very relevant to the web domain, solving important problems in web search by studying edge-to-edge relationships in graph structures. The authors propose the SimEdge model, which improves on SimRank by considering transitivity in similarity measures. The method solves limitations in existing models and is efficient for large-scale graphs.

The approach is innovative, using efficient techniques to calculate edge-to-edge similarities. The paper provides clear mathematical explanations, well-defined algorithms, and theoretical guarantees, such as convergence and error bounds, making the work strong technically.

The scalability of the model is proven through experiments on large datasets, with results evaluated using popular metrics like MAP and NDCG. Overall, this work is a good contribution, combining a practical solution with solid theory and effective experimental validation.

**Questions:**

1. Consider adjusting the layout of Figure 1 after the definition of Example 1 to improve readability.
2. Combine Figure 2 and Figure 3 might result in a more concise presentation.
3. In line 340, it would be helpful to explain the meaning of n and m in the context of the forward incidence matrix.
4. In lines 405 and 414, please add a space before the notations for the R and S matrices.
5. In line 435, please add a space before the notation for the X matrix.

**Reviewer Confidence:**

3: The reviewer is confident but not certain that the evaluation is correct

**Scope:**

4: The work is relevant to the Web and to the track, and is of broad interest to the community

---

### Official Review · Reviewer_nKuA · 2024-12-02

**Novelty:** 5
**Technical Quality:** 5

**Review:**

This paper introduces SimEdge, a scalable and transitivity-aware similarity model designed to assess edge-to-edge relationships in large graphs. Traditional node-to-node similarity models, such as SimRank, often fail to capture transitive relationships or maintain the triangular inequality, limiting their effectiveness for edge-to-edge applications. SimEdge addresses these shortcomings using a mutual reinforcement co-recursion framework that evaluates both node-to-node and edge-to-edge similarities. The model is accompanied by efficient computational methods that scale to graphs with billions of edges. Experimental results demonstrate SimEdge's superiority over existing methods in accuracy, scalability, and meaningfulness of captured relationships.

Strength:

1. The paper focuses on edge-to-edge relationships in graphs, an area that has been relatively neglected compared to node-to-node similarity.

2. SimEdge introduces efficient algorithms with linear memory requirements, which allow it to handle massive graphs with up to 1.4 billion edges. This is a substantial improvement over many existing methods, making it suitable for large-scale applications.

Weakness:

1. While the focus on transitivity is novel, the model primarily builds upon existing ideas like SimRank, extending them rather than completely rethinking the problem. This makes the contribution partially incremental.

2. SimEdge is primarily compared against classical graph similarity models (e.g., SimRank and SimRank*). Modern Graph Neural Networks (GNNs), which could potentially handle edge-to-edge relationships, are not considered as baselines. This omission limits the assessment of SimEdge’s relative novelty in the context of state-of-the-art methods.

**Questions:**

1. While SimEdge emphasizes transitivity, it builds upon existing frameworks like SimRank. Could the authors clarify how SimEdge fundamentally differs from and improves upon SimRank beyond the incorporation of transitivity? What specific use cases would benefit exclusively from SimEdge's features?

2. The paper primarily compares SimEdge with traditional methods (e.g., SimRank). How does SimEdge compare to Graph Neural Networks (GNNs) or other recent deep learning-based approaches that might implicitly capture edge-to-edge relationships? Including such comparisons could provide a broader context for evaluating SimEdge's contributions.

3. The paper mentions SimEdge's scalability, but how does its computational efficiency compare to SimRank or SimRank* on the same datasets in terms of runtime and memory usage?

4. How does SimEdge handle graphs with irregular connectivity patterns or missing data?

**Reviewer Confidence:**

3: The reviewer is confident but not certain that the evaluation is correct

**Scope:**

3: The work is somewhat relevant to the Web and to the track, and is of narrow interest to a sub-community